# Decomposing the Effective Radiative Forcing of anthropogenic aerosols based on CMIP6 Earth System Models

Alkiviadis Kalisoras[1*], Aristeidis K. Georgoulias[1], Dimitris Akritidis[1,2], Robert J. Allen[3], Vaishali Naik[4], Chaincy Kuo[5], Sophie Szopa[6], Pierre Nabat[7], Dirk Olivié[8], Twan van Noije[9], Philippe Le Sager[9], David Neubauer[10], Naga Oshima[11], Jane Mulcahy[12], Larry W. Horowitz[4], Prodromos Zanis[1]

[1]Department of Meteorology and Climatology, School of Geology, Aristotle University of Thessaloniki, Thessaloniki, Greece

[2]Atmospheric Chemistry Department, Max Planck Institute for Chemistry, 55128 Mainz, Germany

[3]Department of Earth and Planetary Sciences, University of California Riverside, Riverside, CA, USA

[4]NOAA Geophysical Fluid Dynamics Laboratory, Princeton, NJ, USA

[5]Lawrence Berkeley National Laboratory, Berkeley, CA 94720, USA

[6]Laboratoire des Sciences du Climat et de l'Environnement, LSCE-IPSL (CEA-CNRS-UVSQ), Université Paris-Saclay, 91191 Gif-sur-Yvette, France

[7]CNRM, Université de Toulouse, Météo-France, CNRS, Toulouse, France

[8]Norwegian Meteorological Institute, Oslo, Norway

[9]Royal Netherlands Meteorological Institute (KNMI), De Bilt, the Netherlands

[10]Institute of Atmospheric and Climate Science, ETH Zurich, Zurich, Switzerland

[11]Meteorological Research Institute, Japan Meteorological Agency, Tsukuba, Japan

[12]Met Office Hadley Centre, Exeter, UK

*Correspondence to*: Alkiviadis Kalisoras (kalisort@geo.auth.gr)

**Abstract.** Anthropogenic aerosols play a major role for the Earth-Atmosphere system by influencing the Earth's radiative budget and precipitation and consequently the climate. The perturbation induced by changes in anthropogenic aerosols on the Earth's energy balance is quantified in terms of the effective radiative forcing (ERF). In this work, the present-day shortwave (SW), longwave (LW), and total (i.e., SW plus LW) ERF of anthropogenic aerosols is quantified using two different sets of experiments with prescribed sea surface temperatures from Earth system models (ESMs) participating in the Coupled Model Intercomparison Project Phase 6 (CMIP6): (a) time-slice pre-industrial perturbation simulations with fixed SSTs (piClim), and (b) transient historical simulations with time evolving SSTs (histSST) over the historical period (1850-2014). ERF is decomposed into three components for both piClim and histSST experiments: (a) $ERF_{ARI}$, representing aerosol-radiation interactions, (b) $ERF_{ACI}$, accounting for aerosol-cloud interactions (including the semi-direct effect), and (c) $ERF_{ALB}$, which is due to temperature, humidity and surface albedo changes caused by anthropogenic aerosols. We present spatial patterns at the top-of-atmosphere and global weighted field means along with inter-model variability (one standard deviation) for all SW, LW, and total ERF components ($ERF_{ARI}$, $ERF_{ACI}$ and $ERF_{ALB}$) and every experiment used in this study. Moreover, the inter-model agreement and the robustness of our results are assessed using a comprehensive method as utilized in the 6[th] IPCC Assessment Report. Based on piClim experiments, the total present-day (2014) ERF from anthropogenic aerosol and precursor emissions is estimated to be $-1.11 \pm 0.26$ W m$^{-2}$, mostly due to the large contribution of $ERF_{ACI}$ in the global mean and the inter-model variability. Based on the histSST experiments for the recent present-day period (1995-2014), similar results are derived, with a global mean total aerosol ERF of $-1.28 \pm 0.37$ W m$^{-2}$ and dominating contribution from $ERF_{ACI}$. The spatial patterns for total ERF and its components are similar in both piClim and histSST experiments. Furthermore, implementing a novel approach to determine geographically the driving factor of ERF, we show that $ERF_{ACI}$ dominates over the largest part of the Earth, $ERF_{ALB}$ mainly over the poles, while $ERF_{ARI}$ over certain reflective surfaces. Analysis of the inter-model variability of total aerosol ERF shows that SW $ERF_{ACI}$ is the main source of uncertainty predominantly over land regions with significant changes in aerosol optical depth (AOD), with East Asia contributing mostly to the inter-model spread of both $ERF_{ARI}$ and $ERF_{ACI}$. The global spatial patterns of total ERF and its components from individual aerosol species, such as sulphates, organic carbon (OC), and black carbon (BC), are also calculated based on piClim experiments. The total ERF caused by sulphates (piClim-SO$_2$) is estimated at $-1.11 \pm 0.31$ W m$^{-2}$, the OC ERF (piClim-OC) is $-0.35 \pm 0.21$ W m$^{-2}$, while the ERF due to BC (piClim-BC) is $0.19 \pm 0.18$ W m$^{-2}$. For sulphates and OC perturbation experiments, $ERF_{ACI}$ dominates over the globe, whereas for BC perturbation experiments $ERF_{ARI}$ dominates over land in the Northern Hemisphere, and especially the

Arctic. Generally, sulphates dominate ERF spatial patterns, exerting a strongly negative ERF especially over industrialized regions of the Northern Hemisphere, such as North America, Europe, East and South Asia. Our analysis on the temporal evolution of ERF over the historical period (1850-2014) reveals that $ERF_{ACI}$ clearly dominates over $ERF_{ARI}$ and $ERF_{ALB}$ for driving the total ERF temporal evolution. Moreover, since the mid-1980s total ERF has become less negative over Eastern North America and Western and Central Europe, while over East and South Asia there is a steady increase in ERF magnitude towards more negative values until 2014.

# 1. Introduction

Anthropogenic aerosols are suspended particles with radii ranging from a few nanometers to a few micrometers (Myhre et al., 2013; Bellouin et al., 2020; Gulev et al., 2021) that are spatially heterogeneously distributed in the atmosphere due to their relatively short lifetime (Lund et al., 2018b; Szopa et al., 2021). Aerosols modify the Earth's radiative budget through direct and indirect processes. Directly, they scatter and absorb incoming solar shortwave (SW) and, to a lesser extent they absorb, scatter and re-emit terrestrial longwave (LW) radiation (Boucher et al., 2013; Bellouin et al., 2020). These processes are denoted as aerosol-radiation interactions (ARI). The net total radiative effect of anthropogenic aerosols partially masks the radiative effect of well-mixed greenhouse gases by cooling the atmosphere (Ming and Ramaswamy, 2009; Szopa et al., 2021); however, where the absorbing aerosol fraction is high they may exert substantial atmospheric warming (Li et al., 2022). Indirectly, tropospheric aerosols alter the radiative and microphysical properties of clouds affecting their reflectivity (or albedo), lifetime, and size, as aerosols can serve as cloud condensation nuclei (CCN) for cloud droplets and ice nucleating particles (INPs) for ice crystals (Haywood and Boucher, 2000; Lohmann and Feichter, 2005; Boucher et al., 2013; Rosenfeld et al., 2014; Bellouin et al., 2020). These processes are denoted as aerosol-cloud interactions (ACI). The aerosol indirect effect is typically divided into two effects. The first indirect effect, also known as cloud albedo effect or Twomey effect, suggests that increased aerosol concentrations in the atmosphere cause increases in droplet concentration and cloud optical thickness due to the presence of more available CCN, with a subsequent decrease in droplet size and an increase of cloud albedo (Twomey, 1974, 1977). The second indirect effect, more commonly known as cloud lifetime effect or Albrecht effect, proposes that a reduction in cloud droplet size due to increased aerosol concentrations affects precipitation efficiency, with a tendency to increase liquid water content, cloud lifetime (Albrecht, 1989), and cloud thickness (Pincus and Baker, 1994). In addition, a semi-direct effect of aerosols can be observed. The term "semi-direct effect" usually refers to the atmospheric heating, with a consequent reduction of relative humidity and therefore cloud amount (i.e., cloud evaporation or cloud burn–off), induced by aerosol absorption locally (Hansen et al., 1997; Ackerman et al., 2000; Allen and Sherwood, 2010). When absorbing aerosols reside above or below clouds then they may enhance cloud cover under some circumstances (Koch and Del Genio, 2010). Nevertheless, in a more general sense, the term semi-direct effect can be used to express the thermodynamic effect of absorbing aerosols on meteorological conditions (atmospheric pressure, temperature profile and cloudiness, etc.) (Tsikerdekis et al., 2019).

The intensities of the direct, semi-direct and indirect effects of aerosols differ among aerosol species. These effects may interact with each other and with other local, regional or global processes, complicating their impacts on precipitation and clouds (Bartlett et al., 2018). Anthropogenic aerosols predominantly scatter SW radiation (Myhre et al., 2013) and produce a net cooling effect globally (Liu et al., 2018). More specifically, sulphate ($SO_4$) particles strongly scatter incoming solar radiation, thus increasing the Earth's albedo and cooling the surface. Sulphate particles also act as CCN, nucleating additional cloud droplets under supersaturated conditions, a process that increases cloud albedo and again has a cooling effect on the Earth-Atmosphere system (Wild, 2009, 2012; Kasoar et al., 2016). Organic aerosols (OAs) generally reflect SW radiation, whereas black carbon (BC) is the most absorbing aerosol particle and strongly absorbs light at all visible wavelengths (Bond et al., 2013; Myhre et al., 2013). Although BC and organic carbon (OC) are co-emitted and have quite similar atmospheric lifetimes, OC scatters sunlight to a much greater degree than BC, thus cooling the atmosphere-surface system (Boucher et al., 2013; Hodnebrog et al., 2016). On the other hand, BC directly absorbs sunlight, heating the surrounding air and reducing the amount of sunlight that reaches the Earth's surface and is reflected back to space (Chen et al., 2010; Bond et al., 2013). Furthermore, when BC is located above a reflective surface, such as snow or clouds, it absorbs the solar radiation reflected from that surface, a process with potentially significant effect over the Arctic (Sand et al., 2013; Stjern et al., 2019). Black carbon interactions with solar radiation depend on its altitude within the troposphere, its position relative to clouds, and the type of the underlying surface (Ramanathan and Carmichael, 2008; Bond et al., 2013).

The aerosol effects discussed above are competing and the calculation of the forcing that aerosols exert on the Earth's climate includes many uncertainties. Difficulties in modeling the radiative forcing of aerosols arise from their complex nature, as their chemical composition and size distribution can rapidly change, and also from the complicated interactions between aerosols, radiation and clouds (Bauer et al., 2020). Climate models lack the resolution to capture small-scale processes that affect the hygroscopic growth of aerosols and the amount of light scattered by them (uncertainties in ARI), and coarsely

parameterize clouds and precipitation, and inaccurately represent turbulent mixing (leading to uncertainties in ACI) (e.g., Neubauer et al., 2014), along with many imperfectly known parameters remaining unresolved (Bellouin et al., 2020). Additionally, aerosol emissions and their evolution over the course of time, which influence their spatiotemporal atmospheric distribution, are still large sources of uncertainty (Bauer et al., 2020). Although Earth system models (ESMs) participating in the Coupled Model Intercomparison Project Phase 6 (CMIP6; Eyring et al., 2016) have increased their level of sophistication regarding processes that drive ACI (Meehl et al., 2020; Gliß et al., 2021), their representation of ACI remains a challenge, because of limitations in their representation of significant sub-grid scale processes (Bellouin et al., 2020; Forster et al., 2021). The Sixth Assessment Report (AR6) of the WGI of the Intergovernmental Panel on Climate Change (IPCC) states that a) aerosol interactions with mixed-phase, (deep) convective, and ice clouds, b) contributions from aerosols serving as INPs to radiative forcing, and c) adjustments in liquid water path and cloud cover in response to perturbations caused by aerosols are major sources of uncertainty in ACI simulated by climate models (Forster et al., 2021). Diversity in the representation of aerosol emissions, atmospheric transport, horizontal and vertical distributions, production rates, atmospheric removal processes, optical properties, hygroscopicity, ability to act as CCN or INPs, chemical composition, ageing, mixing state and morphology (Samset et al., 2013; Kristiansen et al., 2016; Peng et al., 2016; Wang et al., 2016; Zanatta et al., 2016; Myhre et al., 2017; Lund et al., 2018a, b; Allen et al., 2019; Yang et al., 2019; Zelinka et al., 2020; Brown et al., 2021; Gliß et al., 2021; Szopa et al., 2021) affect ARI and ACI, with consequent effects on aerosol radiative forcing calculations (Ghan et al., 2016; Forster et al., 2021). Moreover, the magnitude of the radiative forcing due to ACI could also depend on dynamic backgrounds (Zhang et al., 2016) as well as large-scale circulation adjustments (Dagan et al., 2023).

Radiative forcing offers a metric for quantifying how human activities and natural agents alter the energy flow into and out of the Earth's climate system (Ramaswamy et al., 2019). The Effective Radiative Forcing (ERF; measured in W m$^{-2}$) was recommended as a metric of climate change in the IPCC Fifth Assessment Report (AR5) WGI (Boucher et al., 2013; Myhre et al., 2013) and quantifies the energy that is gained or lost by the Earth-Atmosphere system after an imposed perturbation, rendering it a basic driver of changes in the top-of-the-atmosphere (TOA) energy budget of Earth (Forster et al., 2021). The total ERF due to anthropogenic aerosols over the industrial era (1750-2011) in AR5 was estimated at -0.9 (-1.9 to -0.1) W m$^{-2}$ (uncertainty values in parentheses represent the 5-95% confidence range), with the ERF due to aerosol-radiation interactions (ERF$_{ARI}$) being -0.45 (-0.95 to 0.05) W m$^{-2}$ and the ERF caused by aerosol-cloud interactions (ERF$_{ACI}$) being -0.45 (-1.2 to 0.0) W m$^{-2}$ (Boucher et al., 2013; Myhre et al., 2013). It should be stressed that in AR5, ERF$_{ACI}$ was defined as ERF$_{ARI+ACI}$ minus ERF$_{ARI}$ (Myhre et al., 2013). Since AR5 there have been improvements in ERF estimation due to greater process-understanding and advances in observational and modelling analyses, which have led to an increase in the estimated total aerosol ERF magnitude, along with a reduction in its uncertainty (Forster et al., 2021). As reported in AR6, the total ERF due to aerosols is estimated at -1.3 (-2.0 to -0.6) W m$^{-2}$ over the industrial era (1750–2014), with ERF$_{ARI}$ being estimated at -0.3 (-0.6 to 0.0) W m$^{-2}$ and ERF$_{ACI}$ having a value of -1.0 (-1.7 to -0.3) W m$^{-2}$ (Forster et al., 2021). It should be noted that there remains substantial uncertainty concerning the adjustment contribution to ERF$_{ACI}$ and processes not represented by current ESMs (particularly the effects of aerosols on convective, mixed-phase and ice clouds) (Forster et al., 2021).

A number of recent studies examined the ERF that aerosols exert on the climate system using simulations from CMIP6 models (summarized in Table 1). However, there are several gaps as in many cases their results are based on a single model (e.g., Michou et al., 2020; Oshima et al., 2020), in other cases the ERF patterns are missing (e.g., Thornhill et al., 2021), while in some studies ERF is not further decomposed (e.g., Zanis et al., 2020). This study fills those gaps as well as builds on existing studies by analyzing the spatial and temporal variability of ERF from a multi-model ensemble, comprised of seven CMIP6 ESMs that produced all diagnostics needed to implement the ERF decomposition method proposed by Ghan (2013). The present-day anthropogenic aerosol ERF is examined at the top-of-atmosphere using two different sets of experiments with fixed sea surface temperatures (SSTs) and sea ice cover (SIC) for comparison purposes. Moreover, the evolution of transient ERF during the historical period (1850-2014) is investigated globally and over certain emission regions of the Northern Hemisphere (NH), focusing on the last 20 years of the historical period (1995-2014) in order to mitigate the effects of the negative ERF peak around in late 1970s (Szopa et al., 2021). Apart from the full decomposition of ERF into its ARI, ACI and ALB (Ghan's other forcing term; see Section 2.3) components for all the aerosols and each anthropogenic sub-type separately (SO$_4$, OC, BC), the robustness of ERF results is calculated with a new method based on their statistical significance and inter-model agreement on the sign of ERF. Additionally, the relative contribution of each ERF component geographically is also presented using a novel approach to our knowledge. In brief, this paper is structured as follows. Details about the CMIP6 ESMs and the corresponding simulations used, along with a description of the applied methodology are given in Section 2. The results of this study are presented, discussed, and compared with the results of other studies in Section 3, while at the end of the paper (Section 4) the main conclusions of this research are summarized.

# 2. Data and Methodology

## 2.1 Models Description

The ERF of anthropogenic aerosols was estimated using simulations from seven different ESMs (Table 2) carried out within the framework of RFMIP (Pincus et al., 2016) and AerChemMIP (Collins et al., 2017), which were endorsed by CMIP6 (Eyring et al., 2016). Anthropogenic emissions of aerosols as well as aerosol and ozone precursors (excluding methane) used by climate models are from van Marle et al. (2017) and Hoesly et al. (2018), while each model uses its own natural emissions (Eyring et al., 2016).

The CNRM-ESM2-1 model (Séférian et al., 2019; Michou et al., 2020) uses the Reactive Processes Ruling the Ozone Budget in the Stratosphere Version 2 (REPRO-BUS-C_v2) atmospheric chemistry scheme, in which chemical evolution is calculated only above the 560 hPa level. Below that level, the concentrations of the species are relaxed either toward the yearly evolving global mean abundances (Meinshausen et al., 2017) or toward the 560-hPa value. The Tropospheric Aerosols for ClimaTe In CNRM (TACTIC_v2) interactive tropospheric aerosol scheme is also used in CNRM-ESM2-1, which implements

a sectional representation of BC, organic matter, sulphates, sea-salt and desert dust. The $SO_4$ precursors evolve in sulphate aerosols with dependence on latitude (Séférian et al., 2019). The cloud droplet number concentration is dependent on the concentrations of sea-salt, sulphate and organic matter, thus representing the cloud albedo (or Twomey) effect, but not any other aerosol-cloud effects.

     EC-Earth3-AerChem (van Noije et al., 2021) is an extended version of EC-Earth3 (Döscher et al., 2022) that can

simulate tropospheric aerosols, methane, ozone and atmospheric chemistry. It utilizes the McRad radiation package, which includes a SW and LW radiation scheme that is based on the Rapid Radiative Transfer Model for General Circulation Models (RRTMG) (van Noije et al., 2021). It treats the radiative transfer in clouds using the Monte Carlo independent column approximation (McICA) (Morcrette et al., 2008). Atmospheric chemistry and aerosols are simulated with the Tracer Model version 5 release 3.0 (TM5-mp 3.0), which includes sulphate, black carbon, organic aerosols, mineral dust and sea salt (van

Noije et al., 2021). The modal aerosol microphysical scheme M7 (Vignati et al., 2004) describes the aforementioned aerosol species and is made up of four water-soluble modes (nucleation, Aitken, accumulation and coarse) and three insoluble modes (Aitken, accumulation and coarse), with each mode being described by a lognormal size distribution that has a fixed geometric standard deviation. M7 describes the evolution of total mass and particle number of each species for each mode and accounts for water uptake, new particle formation and aging through coalescence and condensation (Vignati et al., 2004).

The GFDL-ESM4 model (Dunne et al., 2020; Horowitz et al., 2020) consists of the Geophysical Fluid Dynamics Laboratory (GFDL)'s Atmosphere Model version 4.1 (AM4.1), which includes an interactive tropospheric and stratospheric gas-phase and aerosol chemistry scheme. In contrast to the previous model version (AM4.0), nitrate and ammonium aerosols are treated explicitly, the rate of aging of BC and OC from hydrophobic to hydrophilic forms changes depending on the calculated concentrations of hydroxyl radical (OH), and oxidation of $SO_2$ and dimethyl sulfide to produce sulphate aerosols is

driven by the gas-phase oxidant concentrations (OH, ozone and $H_2O_2$) and cloud pH (Horowitz et al., 2020). Aerosols are represented as bulk concentrations of sulfate, nitrate, ammonium, and hydrophilic and hydrophobic BC and OC, plus five bins each for sea salt and mineral dust. Sulphate and hydrophilic black carbon aerosols are considered to be internally mixed by the radiation code.

     The MPI-ESM-1-2-HAM model (Lohmann and Neubauer, 2018; Mauritsen et al., 2019; Neubauer et al., 2019e;

Tegen et al., 2019) is the latest version of the Max Planck Institute for Meteorology Earth System Model (MPI-ESM1.2) coupled with the Hamburg Aerosol Model version 2.3 (HAM2.3). It contains the atmospheric general circulation model ECHAM6.3 developed by the Max Planck Institute for Meteorology. ECHAM6.3–HAM2.3 uses a two-moment cloud microphysics scheme to study aerosol–cloud interactions and improve the simulation of clouds. The aerosol–cloud interactions are simulated in liquid, mixed-phase and ice clouds (Neubauer et al., 2019e). The aerosol microphysics module HAM

calculates the evolution of aerosol particles considering the species BC, OC, sulphate, sea salt and mineral dust. In its default version, HAM simulates the aerosol spectrum as the superposition of 7 lognormal modes (nucleation, Aitken, accumulation and coarse modes) (Tegen et al., 2019). For aerosol activation the scheme by Abdul-Razzak and Ghan (2000) (implemented by Stier, 2016) and for autoconversion of cloud droplets to rain the scheme by Khairoutdinov and Kogan (2000) are used.

     The MRI-ESM2 model (Kawai et al., 2019; Yukimoto et al., 2019f; Oshima et al., 2020) includes the MRI Chemistry

Climate Model version 2.1 (MRI-CCM2.1) atmospheric chemistry model, which computes the evolution and distribution of ozone and other trace gases in the troposphere and middle atmosphere, and the Model of Aerosol Species in the Global Atmosphere mark-2 revision 4-climate (MASINGAR mk-2r4c) aerosol model, which contains BC, OC, non sea-salt sulphate, mineral dust, sea salt and aerosol precursor gases (e.g., $SO_2$ and dimethyl sulfide), assuming external mixing for all aerosol species (Yukimoto et al., 2019f). In MASINGAR mk-2r4c the conversion rate of hydrophobic to hydrophilic BC is depended

on the rate at which condensable materials cover hydrophobic BC, an approach that could reproduce the seasonal variations of BC mass concentrations that are observed over the Arctic region (Oshima et al., 2020).

     NorESM2-LM (Kirkevåg et al., 2018; Seland et al., 2020) is the "low resolution" version of the second version of the coupled Norwegian Earth System Model (NorESM2). It employs the CAM6-Nor atmosphere model, which uses

parameterization schemes for aerosols and aerosol–radiation–cloud interactions, and the OsloAero6 atmospheric aerosol module, which describes the formation and evolution of BC, OC, sulphate, dust, sea salt and secondary organic aerosol. The oxidant concentrations of OH, ozone, $NO_3$ and $HO_2$ are prescribed by 3D monthly mean fields (Seland et al., 2020).

The UKESM1 model (Sellar et al., 2020) uses the U.K. Chemistry and Aerosol (UKCA) interactive stratosphere–troposphere chemistry scheme (UKCA StratTrop) (Archibald et al., 2020) and the GLOMAP microphysical aerosol scheme Mann et al. (2010). GLOMAP is a two-moment modal aerosol microphysics scheme that simulates the sources, evolution and sinks of black carbon, sulphates, organic matter and sea salt across five lognormal size modes (Mulcahy et al., 2020). Mineral dust is simulated independently using the CLASSIC dust scheme (Bellouin et al., 2011) and can, therefore, be considered to be externally mixed with the aerosols of GLOMAP (Mulcahy et al., 2020).

## 2.2 CMIP6 Simulations Description

To quantify the pre-industrial to present-day ERF due to anthropogenic aerosols, ESMs that performed time-slice experiments (Table 3) covering a period of at least 30 years of simulation with a fixed monthly averaged climatology of SSTs and SIC corresponding to the year 1850 were used. Each model performed five time-slice experiments: one control experiment (piClim-control) and four perturbation experiments (piClim-aer, piClim-BC, piClim-OC, and piClim-$SO_2$). Albeit not truly pre-industrial, the year 1850 is considered as a pre-industrial period in an attempt to create a stable near-equilibrium climate state that represents the period before the beginning of large-scale industrialization (Eyring et al., 2016). The number of simulation years chosen for the aforementioned experiments is the minimum value in order to account for internal variability, which generates substantial interannual variability in the ERF estimates (Collins et al., 2017), and to constrain global forcing to within 0.1 W m$^{-2}$ (Forster et al., 2016). In cases where simulations were longer than 30 years, only the final 30-year period was chosen. The piClim-control simulation uses fixed 1850 values for concentrations of well-mixed greenhouse gases including $CO_2$, methane, nitrous oxide, aerosols and aerosol precursors, ozone precursors and halocarbon emissions or concentrations, and land use and solar irradiance. Each perturbation simulation is run similarly for the same 30-year period as the control simulation, keeping the SSTs and SIC fixed to pre-industrial levels (1850), but setting one or more of the specified species (concentrations or emissions) to present-day (2014) values (Collins et al., 2017). Consequently, piClim-BC, piClim-OC, and piClim-$SO_2$ experiments, use precursor emissions of 2014 for BC, OC, and $SO_2$ (which is the precursor of sulphates), respectively, while all other forcings are set to 1850 values. In the piClim-aer simulation, all anthropogenic aerosol precursor emissions are set to 2014 values with all other forcings set to 1850 values.

In order to calculate the transient aerosol ERF over the historical period, ESMs which performed transient historical experiments for the period between 1850 and 2014 with prescribed SSTs and sea ice were considered. The histSST and histSST-piAer experiments share the same forcings as the "historical" experiment (see also Eyring et al., 2016) and both use the monthly mean time-evolving SST and sea ice values from one ensemble member of the historical simulations (the same SSTs and sea ice values are used for both the control and perturbation experiments), but the latter uses aerosol precursor emissions of the year 1850 (Collins et al., 2017). While this is technically not an ERF (since SSTs and SIC are evolving), the impact of transient SSTs and sea ice on ERF diagnosis is considered to be small (Forster et al., 2016; Collins et al., 2017). For the purpose of comparing the present-day ERF of anthropogenic aerosols between the piClim and the histSST experiments, the last 20 years of the historical period (1995-2014) were chosen because it is the most recent period available in CMIP6 histSST simulations while mitigating the effects of the negative ERF peak around 1980 (Szopa et al., 2021). We performed this comparison to show the consistency between the all-anthropogenic-aerosol ERFs calculated using two different sets of experiments.

## 2.3 Methodology

ERF is considered as the change in net downward TOA radiative flux after allowing both tropospheric and stratospheric temperatures, water vapor, clouds, and some surface properties that are not coupled to any global surface air temperature change to adjust (Smith et al., 2018; Forster et al., 2021). By fixing SSTs and SIC at climatological values, all other parts of the system are allowed to respond until reaching steady state (Hansen, 2005). This allows for ERF to be diagnosed as the difference in the net flux at the TOA between the perturbed experiments and the control simulation (Hansen, 2005; Sherwood et al., 2015). The fixed-SST method is less sensitive to internal climate variability as it benefits from the long averaging times and the absence of interannual ocean variability in the perturbed and control simulations (Sherwood et al., 2015), and can reduce the 5–95% confidence range of ERF estimations up to 0.1 W m$^{-2}$ (Forster et al., 2016). The ERF of anthropogenic aerosols was analyzed here following the method of Ghan (2013), which is also known as the "double call" method, meaning that the ESM radiative flux diagnostics are calculated a second time neglecting aerosol scattering and absorption (Ghan, 2013). In order to distinguish and quantify the magnitude of different processes to the total ERF, the effective radiative forcing was split into three main components: (a) ERF$_{ARI}$, which represents the aerosol-radiation interactions (i.e.,

scattering and absorption of radiation by aerosol particles; Eq. 1), (b) ERF$_{ACI}$, which accounts for all changes in clouds and aerosol-cloud interactions (i.e., the effects of aerosols on cloud radiative forcing; Eq. 2), and (c) ERF$_{ALB}$, which is Ghan's other forcing term and is mostly the contribution of surface albedo changes in the SW that are caused by aerosols (Eq. 3) (Ghan, 2013). Consequently, the sum of ERF$_{ARI}$, ERF$_{ACI}$, and ERF$_{ALB}$ gives an approximation of the overall ERF of aerosol species (Eq. 4):

$$ERF_{ARI} = \Delta (F - F_{af}), \qquad (1)$$
$$ERF_{ACI} = \Delta (F_{af} - F_{csaf}), \qquad (2)$$
$$ERF_{ALB} = \Delta F_{csaf}, \qquad (3)$$
$$ERF = ERF_{ARI} + ERF_{ACI} + ERF_{ALB}, \qquad (4)$$

where F is the net (downward minus upward) radiative flux at the TOA, $F_{af}$ (af: aerosol-free) is the flux calculated ignoring the scattering and absorption by aerosols, despite their presence in the atmosphere (i.e., aerosol-free forcing), $F_{csaf}$ (csaf: clear-sky, aerosol-free) is the flux calculated neglecting the scattering and absorption by both aerosols and clouds, and $\Delta$ denotes the difference between the perturbation and the control experiment. The ERF$_{ACI}$ term is an estimate of anthropogenic aerosol effects on cloud radiative forcing, which is the sum of aerosol indirect effects and semi-direct effects (Ghan et al., 2012; Ghan, 2013; Zelinka et al., 2023b). The term ERF$_{ALB}$ is not only influenced by aerosol-induced changes in surface albedo (Zelinka et al., 2023b), but it is used here for compatibility purposes with the respective term used in the paper of Ghan (2013). As shown in the work of Zelinka et al. (2023) the SW ERF$_{ALB}$ includes the change in net radiation caused by surface albedo changes, the aerosol-free clear-sky radiative contributions from humidity changes, and a masking term which represents the radiative impact of surface albedo changes that is attenuated by the presence of both aerosols and clouds. On the other hand, the LW component of ERF$_{ALB}$ includes the aerosol-free clear-sky radiative contributions from changes in temperature and humidity (Zelinka et al., 2023b).

In this work, piClim-control was subtracted from piClim-aer, piClim-BC, piClim-OC, and piClim-SO$_2$ in order to calculate the present-day anthropogenic aerosol ERF (from all aerosols, BC, OC, and SO4, respectively) on a global scale, and histSST-piAer was subtracted from histSST to estimate the transient anthropogenic aerosol ERF during the 1995-2014 period. Moreover, the time evolution of the total ERF and its decomposition into ERF$_{ARI}$, ERF$_{ACI}$, and ERF$_{ALB}$ during the historical period (1850-2014) was examined globally and over certain reference regions. The approach described above was implemented for both the SW and LW radiation, with their sum providing an estimation of the total ERF for each component (Eq. 5-8):

$$ERF_{ARI\,(TOTAL)} = ERF_{ARI\,(SW)} + ERF_{ARI\,(LW)}, \qquad (5)$$
$$ERF_{ACI\,(TOTAL)} = ERF_{ACI\,(SW)} + ERF_{ACI\,(LW)}, \qquad (6)$$
$$ERF_{ALB\,(TOTAL)} = ERF_{ALB\,(SW)} + ERF_{ALB\,(LW)}, \qquad (7)$$
$$ERF_{TOTAL} = ERF_{ARI\,(TOTAL)} + ERF_{ACI\,(TOTAL)} + ERF_{ALB\,(TOTAL)}. \qquad (8)$$

Due to differences in the spatial horizontal resolution of the ESMs (Table 2), all data were regridded to a common spatial grid (2.8125º x 2.8125º) by applying bilinear interpolation prior to processing. Due to lack of aerosol-free diagnostics (see Table A1 in Appendix A for the description of the CMIP6 variables used in this study), EC-Earth3-AerChem was not included in the piClim-BC, piClim-OC and piClim-SO$_2$ analysis, while MRI-ESM2-0 was not included in the histSST analysis. Along with ERF, the differences in aerosol optical depth (AOD) at 550 nm due to present-day anthropogenic aerosols were also calculated for both the piClim and histSST sets of experiments for comparison purposes. The statistical significance of both ERF and $\Delta$AOD results was tested at the 95% confidence level using a paired sample t-test that was conducted to the results of each model. The robustness of the multi-model ensemble results in Figs. 1-3 was estimated based on the statistical significance of each model's results as well as the agreement on the sign of change between ESMs. The exact criteria for determining the robustness of the results are described in Table A2 within Appendix A.

# 3. Results

## 3.1 AOD changes in piClim and histSST experiments

The magnitude of ERF is affected by aerosol concentrations in the atmosphere. Thus, the differences in pre-industrial to present-day ambient aerosol optical depth ($\Delta$AOD) at 550 nm due to aerosols are presented in Fig. 1, serving as an indicator of the amount of aerosols in the atmosphere. AOD is the column-integrated measure of solar intensity extinction caused by aerosols at a given wavelength being also related to aerosol mass concentrations (Szopa et al., 2021). The multi-model annual mean $\Delta$AOD between piClim-aer and piClim-control simulations (which represents the change in AOD over the 1850-2014

period) is 0.0299 ± 0.0082 (all ranges are given as one standard deviation across models), a value that is very close to the mean annual difference between histSST and histSST-piAer for the period 1995-2014 (the period closest to the end of historical; hereafter denoted as EHP), which is calculated to be 0.0302 ± 0.0088. The ΔAOD for all aerosols is positive over most of the globe, with the highest values found primarily over South and East Asia, and secondarily over Indonesia, Europe, and Eastern United States (Fig. 1a, b). Four ΔAOD regimes can be distinguished: a) high to medium ΔAOD over land (East and South Asia, Eastern Europe, Middle East and Eastern North America), b) medium to low ΔAOD over land (North and South America, Western Europe, Greenland, Oceania, Antarctica and the Arctic), c) high to medium ΔAOD over ocean (Northwestern Pacific and Northernmost Indian), and d) medium to low ΔAOD over ocean (Atlantic, South Pacific and South Indian).

Spatial distribution of the ambient aerosol ΔAOD is notably influenced by the pattern of sulphates, with the mean global $SO_4$ AOD difference being 0.0191 ± 0.0057 and 0.0191 ± 0.0077 for the piClim and histSST experiment sets, respectively, which is almost equal to two-thirds of the ambient aerosol AOD difference (Fig. 1c, d). Organic aerosols exhibit quite a different pattern than sulphates, as their peak positive AOD differences are confined to biomass burning regions. The global mean AOD difference between piClim-OC and piClim-control is 0.0046 ± 0.0011 and 0.0073 ± 0.0039 between that of histSST and histSST-piAer corresponding to EHP (Fig. 1e, f). The highest positive changes between pre-industrial and present-day AOD for black carbon are over East and South Asia, with an annual global value of 0.0040 ± 0.0018 for the piClim experiments and 0.0018 ± 0.0005 for the EHP in histSST experiments (Fig. 1g, h). Note that the AOD changes for sulphates (Fig. 1d), organic aerosols (Fig. 1f), and black carbon (Fig. 1h) were calculated only for a subset of models (CNRM-ESM2-1, EC-Earth3-AerChem, GFDL-ESM4, and NorESM2-LM), which were the only ones that provided the necessary CMIP6 variables (od550so4, od550oa, od550bc, respectively; Table A1). The global mean values of AOD changes for each model and each experiment can be found in Table S1 in the electronic supplement.

## 3.2 Decomposition of ERF for all anthropogenic aerosols

Following Ghan (2013), the TOA radiative flux difference between the control and perturbation simulations in both shortwave (SW) and longwave (LW) was calculated for each of the models to estimate the total (SW+LW) aerosol ERF. The multi-model global mean values for the total ERF and its decomposition into $ERF_{ARI}$, $ERF_{ACI}$ and $ERF_{ALB}$ are presented in Table 4 as well as in Figs. 2-5. The global mean values of SW and LW ERF for each model and each experiment are provided in Tables S2-S4, while the SW and LW ERF patterns at TOA for the multi-model ensemble are shown in Figs. S1 and S2 in the electronic supplement, respectively.

As seen in Fig. 2, the global mean ERF due to pre-industrial to present day changes in all anthropogenic aerosols is -1.11 ± 0.26 W m$^{-2}$, while the mean total ERF value during EHP is calculated to be -1.28 ± 0.37 W m$^{-2}$ (Fig. 2a, b). There are small differences in ERF for EHP calculated in this study versus that shown by Szopa et al. (2021), which are related to a weighting issue in the global mean net ERF in Fig 6.11 in IPCC WGI AR6 Chapter 6 that accounts for an excess of -0.25 W m$^{-2}$ in the peak. The authors of Szopa et al. (2021) are working to remedy this record. In the current analysis, the global mean total ERF during that period (1965-1984) is calculated to be -1.27 ± 0.43 W m$^{-2}$. Although there are slight differences over certain regions, a quite common spatial TOA pattern for ERF emerges between piClim-aer and histSST experiments: anthropogenic aerosols induce a negative total ERF over the globe, especially over the NH, with the most negative values mainly over East Asia, followed by South Asia, Europe and North America, while the most positive values are found over reflective continental surfaces, such as the Sahara, Alaska, Greenland and the Arabian Peninsula (Fig. 2a, b). The high surface albedo of the latter regions decreases (increases) the effect of scattering (absorbing) aerosols, thus leading to a positive ERF (Myhre et al., 2013; Shindell et al., 2013; Zanis et al., 2020). The areas with peak negative ERF values are a robust feature among all ESMs included in this study, despite any differences in ERF magnitude (Figs. S3 and S4).

Clearly, $ERF_{ACI}$ dominates the total ERF on a global scale, as it exhibits a pattern almost identical to that of the total ERF (Fig. 2e, f). The multi-model mean $ERF_{ACI}$ in piClim-aer is -1.14 ± 0.33 W m$^{-2}$, while the histSST $ERF_{ACI}$ is estimated at -1.24 ± 0.44 W m$^{-2}$ during EHP. The impact of aerosol-cloud interactions on the total ERF is highlighted, as peak negative $ERF_{ACI}$ regions coincide with the ones of total ERF for both experiments. The mean $ERF_{ARI}$ is slightly negative globally, although not statistically significant, with a mean value of -0.02 ± 0.20 W m$^{-2}$ for piClim-aer and -0.08 ± 0.14 W m$^{-2}$ for histSST experiments. In both cases, peak positive values of $ERF_{ARI}$, which can be attributed to absorbing aerosol particles, are found over parts of Central Africa, the Arabian Desert and continental East Asia, whereas the most negative values are detected over the oceanic regions surrounding India. Interestingly, $ERF_{ARI}$ is positive over the Arctic and Antarctica (Fig. 2c, d). On the other hand, $ERF_{ALB}$ is slightly positive on a global scale and is calculated to be 0.05 ± 0.07 W m$^{-2}$ and 0.04 ± 0.08 W m$^{-2}$ for the piClim-aer and histSST (1995-2014) simulations, respectively. The highest $ERF_{ALB}$ values appear particularly over the Himalayas, and the adjacent regions in South Asia, while mostly negative values are seen over the poles (Fig. 2g, h).

It should be noted that the global mean ERF values show significant differences among the ESMs (Tables S2-S4 and Figs. S3 and S4). The CNRM-ESM2-1 and GFDL-ESM4 models produce the weakest total ERF due to their small $ERF_{ACI}$. The decreased ERF magnitude of GFDL-ESM4 compared with their previous-generation AM3 model can be attributed to a

reduction in the strength of the aerosol indirect effect due to changes in the model's horizontal resolution and modifications in representations of certain aerosol processes (Zhao et al., 2018; Horowitz et al., 2020), while CNRM-ESM2-1 only represents the first indirect (i.e., cloud albedo) effect without the inclusion of any secondary aerosol indirect effects (impacts on precipitation; Michou et al., 2020). On the other hand, EC-Earth3-AerChem, MPI-ESM-1-2-HAM and NorESM2-LM exhibit a strongly negative $ERF_{ACI}$. In the case of MPI-ESM-1-2-HAM, the strongly negative $ERF_{ACI}$ probably results from an overestimation of cloud-top cloud droplet number concentrations, leading to a subsequent overestimation of SW cloud radiative effect in regions where shallow convective clouds are common (Neubauer et al., 2019e). Another reason for the strongly negative $ERF_{ACI}$ in MPI-ESM-1-2-HAM could be the highly negative liquid water path adjustments calculated in ECHAM6.3-HAM2.3 on which MPI-ESM-1-2-HAM is based (Gryspeerdt et al., 2020).

## 3.3 Decomposition of ERF for different anthropogenic aerosol types

To quantify the effect of different aerosol species on the total radiative forcing induced by anthropogenic aerosols, ERF was calculated for piClim-BC, piClim-OC, and piClim-SO$_2$ (there are no equivalent single-aerosol species transient historical simulations with fixed SSTs for comparison) in the same manner as in Section 3.2 (Table 4 and Fig. 3). The global mean values of SW and LW ERF for each model and each aerosol type experiment can be found in Tables S2-S3, while the SW and LW ERF patterns at TOA for the multi-model ensemble are shown in Figs. S5 and S6, respectively. The ERF decomposition for each model for piClim-SO$_2$, piClim-OC, and piClim-BC are presented in Figs. S7-S9, respectively.

There is a pronounced similarity between piClim-aer and piClim-SO$_2$ in both the global means and the spatial TOA pattern of the total ERF (Fig. 3), consistent with the dominant contribution of sulphate AOD to ambient aerosol AOD changes. Sulphate particles highly scatter incoming SW solar radiation, causing a negative ERF over the NH, in general, and over the emission sources (i.e., continental East and South Asia, followed by Europe and N. America) and downwind regions, in particular, thus playing a dominant role in the overall TOA radiative forcing. The global mean total ERF due to SO$_4$ is -1.11 ± 0.31 W m$^{-2}$ (Fig. 3a), nearly equal to the total ERF in the combined-aerosol experiment (piClim-aer). However, there is a larger contribution to the total sulphate ERF from its ARI component, which is almost entirely negative over the globe (Fig. 3d), with peak negative values over East and South Asia, and a global mean value of -0.32 ± 0.12 W m$^{-2}$. Furthermore, sulphate $ERF_{ACI}$ is almost 30% less negative than the respective $ERF_{ACI}$ in piClim-aer, with a multi-model mean value of -0.83 ± 0.23 W m$^{-2}$, peaking over East Asia and driving the bulk of total ERF from SO$_4$ (Fig. 3g). The global mean $ERF_{ALB}$ of piClim-SO$_2$ is 0.03 ± 0.09 W m$^{-2}$ showing a positive peak over the northern part of the Middle East, which is not statistically significant (Fig. 3j).

Organic carbon causes a less negative ERF on the climate system than sulphates, with a global mean value of -0.35 ± 0.21 W m$^{-2}$, which peaks over Southeast Asia (Fig. 3b). $ERF_{ACI}$ is estimated to be -0.27 ± 0.24 W m$^{-2}$ and greatly affects the total ERF pattern (Fig. 3h). Despite having a globally negative mean value, the ERF pattern at TOA due to OC (in piClim-OC) does not resemble that of piClim-SO$_2$ or piClim-aer, which can be attributed to different emission sources and radiative properties (see also Li et al., 2022). For instance, in piClim-OC there is an evident positive ERF over the Eastern United States and West Europe, regions where negative ERF was detected in piClim-aer and piClim-SO$_2$. $ERF_{ARI}$ due to OC is negative over most continental regions (Fig. 3e), with a global mean value of -0.08 ± 0.04 W m$^{-2}$, while the global mean sign of $ERF_{ALB}$ is unclear as the global mean forcing is estimated at 0.01 ± 0.03 W m$^{-2}$ (Fig. 3k).

Black carbon is the most absorbing aerosol species (Myhre et al., 2013) and it strongly absorbs light at all visible wavelengths (Bond et al., 2013), thus inducing a positive ERF at TOA (Ramanathan and Carmichael, 2008). Globally the mean total ERF caused by BC is calculated to be 0.19 ± 0.18 W m$^{-2}$, with pronounced positive peaks over South and East Asia, the Arabian Desert, and Central Africa (Fig. 3c). In contrast to the above piClim perturbation simulations, the spatial distribution of total BC ERF at TOA is principally affected by $ERF_{ARI}$ instead of $ERF_{ACI}$, with the former having a global mean value greater than the total ERF by a factor of nearly two (Table 4). BC $ERF_{ARI}$ is positive all over the globe and has a mean value of 0.39 ± 0.19 W m$^{-2}$, peaking over the same regions as total BC ERF (Fig. 3f), while $ERF_{ACI}$ is -0.20 ± 0.30 W m$^{-2}$ and shows no statistically significant peaks (Fig. 3i). The global mean sign of BC $ERF_{ALB}$ is also not clear, as it is calculated to be 0.00 ± 0.05 W m$^{-2}$, with the most positive (although not statistically significant) values detected over Southern continental Asia (Fig. 3l).

## 3.4 SW and LW contributions to ERF

Investigation of the relative contribution from SW and LW ERFs to the total ERF reveals that the SW component is mainly responsible for the total ERF values calculated using the Ghan (2013) method. In Figs. 4 and 5 the total, SW, and LW values for $ERF_{ARI}$, $ERF_{ACI}$, $ERF_{ALB}$, as well as their sum are shown for the combined-aerosol experiments (Fig. 4) and the single-aerosol-species experiments (Fig. 5). The SW and LW values for all ERF components in every experiment are presented for each model and their ensemble in Tables S2-S4.

In the all-aerosol simulations (piClim-aer and histSST averaged over the EHP), although all SW (LW) ERF components have negative (positive) values, in the cases of $ERF_{ARI}$ and $ERF_{ACI}$ the SW component has higher absolute values than the LW and greatly influences their respective total ERF values, whereas the opposite applies to $ERF_{ALB}$ (Fig. 4). Total $ERF_{ARI}$ exhibits a larger spread among ESMs in piClim (varying from -0.32 W m$^{-2}$ to 0.26 W m$^{-2}$) than in histSST (with values ranging from -0.27 W m$^{-2}$ to 0.08 W m$^{-2}$), whereas the opposite stands for the total $ERF_{ACI}$, with a range between -1.57 W m$^{-2}$ and -0.61 W m$^{-2}$ in piClim-aer, and -1.86 W m$^{-2}$ and -0.59 W m$^{-2}$ in histSST. This shows the similarities between piClim-aer and histSST (averaged over 1995-2014), but also highlights the differences between ESMs in $ERF_{ARI}$ and $ERF_{ACI}$. While all models agree on the negative sign of $ERF_{ACI}$ for both experiments, there are discrepancies in the sign of $ERF_{ARI}$. In piClim-aer $ERF_{ACI}$ shows the largest inter-model variability among the three main ERF components in both the SW (ranging from -2.49 W m$^{-2}$ to -0.59 W m$^{-2}$) and the LW (with a range between -0.17 W m$^{-2}$ and 1.49 W m$^{-2}$) probably owing to different representation of ACI and aerosol microphysical processes among individual ESMs (Bauer et al., 2020; Szopa et al., 2021). GFDL-ESM4, in particular, is the only model with negative total LW ERF (Table S3), whereas MRI-ESM2-0 has the strongest $ERF_{ACI}$ in both the SW (Table S2) and LW (Table S3), with large negative SW $ERF_{ACI}$ and positive LW $ERF_{ACI}$ values caused by the aerosol effects on high-level ice clouds over convective regions in the tropics (Oshima et al., 2020), which eventually cancel each other out in the total $ERF_{ACI}$.

In the histSST experiment (averaged over the EHP) individual ESMs exhibit smaller differences in their $ERF_{ACI}$ estimates (i.e., less inter-model variability; Table S4), with values ranging from -1.78 W m$^{-2}$ to -0.53 W m$^{-2}$ in the SW. Their LW counterparts have slightly positive or negative values, resulting in a near-zero LW $ERF_{ACI}$ (Table S4), in contrast with the more positive LW $ERF_{ACI}$ presented in piClim-aer (Table S3), due to the highly positive LW $ERF_{ACI}$ obtained from MRI-ESM2-0. Contributions from $ERF_{ARI}$ and $ERF_{ALB}$ to the total ERF are much smaller in both the piClim-aer and histSST experiments, with the former having a marginally negative and the latter slightly positive global mean value (Fig. 4). As the total SW (LW) ERF is the sum of the three individual SW (LW) ERF components, the global multi-model mean ERF value is a result of a strongly negative SW radiative forcing being offset by a weaker, but not negligible, positive LW forcing at TOA. The total $ERF_{ARI}$ is predominantly influenced by SW $ERF_{ARI}$ as aerosols interact with the incoming SW radiation through scattering and absorption. It should be borne in mind that not all ESMs agree on the magnitude or even the sign of the individual SW and LW ERF main components (Tables S2-S4) due to uncertainties in the parameterization schemes used in ESMs to describe the way aerosols interact with radiation and clouds.

The SW, LW and total (SW+LW) values for the three main ERF components and their sum for each anthropogenic aerosol type are presented in Fig. 5. In the case of light-scattering aerosols (i.e., sulphates and organic carbon) the strongly negative SW $ERF_{ACI}$ drives the radiative forcing due to ACI, which in turn is mainly responsible for the negative total ERF values. Sulphates induce forcings due to ARI and ACI at TOA that are larger in magnitude than those of OC. The global mean sulphate $ERF_{ARI}$ ($ERF_{ACI}$) is larger than the respective OC $ERF_{ARI}$ ($ERF_{ACI}$) by a factor of 4 (3), although this may not be the case when examining each ESM individually. $ERF_{ARI}$ and $ERF_{ACI}$ due to $SO_4$ range from -0.49 W m$^{-2}$ to -0.19 W m$^{-2}$ and from -1.11 W m$^{-2}$ to -0.51 W m$^{-2}$, respectively, while $ERF_{ARI}$ and $ERF_{ACI}$ caused by OC vary from -0.15 W m$^{-2}$ to -0.02 W m$^{-2}$ and from -0.79 W m$^{-2}$ to -0.06 W m$^{-2}$, respectively (Table 4). All models agree on the negative sign of SW $ERF_{ARI}$ and SW $ERF_{ACI}$ in both the piClim-$SO_2$ and piClim-OC experiments, with global mean values ranging from -0.53 W m$^{-2}$ to -0.20 W m$^{-2}$ for SW $ERF_{ARI}$ and from -1.40 W m$^{-2}$ to -0.51 W m$^{-2}$ for SW $ERF_{ACI}$ in the piClim-$SO_2$ experiment, and values that vary from -0.16 W m$^{-2}$ to -0.04 W m$^{-2}$ for SW $ERF_{ARI}$ and from -0.80 W m$^{-2}$ to -0.07 W m$^{-2}$ for SW $ERF_{ACI}$ in piClim-OC (Table S2). In both experiments LW $ERF_{ARI}$ is extremely small (the multi-model ensemble mean is 0.01 W m$^{-2}$ for piClim-$SO_2$ and 0.00 W m$^{-2}$ for piClim-OC; Table S3), while there is a widespread agreement among ESMs that LW $ERF_{ACI}$ is slightly positive (only GFDL-ESM4 in piClim-OC exhibits a negative $ERF_{ACI}$ of -0.04 W m$^{-2}$; Table S3). Total $ERF_{ALB}$ is slightly positive globally in piClim-$SO_2$ and piClim-OC experiments, with all but two models agreeing on the positive sign of the forcing (NorESM2-LM in piClim-$SO_2$, and MRI-ESM2-0 and NorESM2-LM in piClim-OC have negative $ERF_{ALB}$ mean values; Table 4). There is a general agreement among models for the signs of SW and LW $ERF_{ALB}$ values in both the piClim-$SO_2$ and piClim-OC experiments (Tables S2 and S3).

On the contrary, light-absorbing BC induces a positive total ERF at TOA, with almost equal contribution from the SW and the LW (Fig. 5). Nearly all individual models produce a positive total BC ERF (Table 4) arising from the positive SW ERF due to absorption of solar incoming radiation (Table S2), which is offset by a negative, but weaker, LW ERF (Table S3). MPI-ESM-1-2-HAM is the only model that has a negative total ERF due to BC (Table 4) because SW $ERF_{ARI}$ and SW $ERF_{ACI}$ cancel each other out completely (Table S2), while MRI-ESM2-0 produces a strongly negative SW $ERF_{ACI}$ and a highly positive LW $ERF_{ACI}$ (Tables S2 and S3), which also cancel each other out, ultimately exhibiting a smaller total $ERF_{ACI}$ and a positive total ERF (Table 4). Although there might be quantitative uncertainties in the strongly negative (positive) SW (LW) $ERF_{ACI}$ produced by MRI-ESM2-0, these values could be explained by an increase in the number concentration of ice crystals in high-level clouds that is caused by BC aerosols, especially over convective regions within the tropics (Oshima et al., 2020; Thornhill et al., 2021). The large inter-model spread in SW and LW BC $ERF_{ACI}$ (and total SW and LW BC ERFs consequently) is explained by the above inconsistencies between individual ESMs. Total $ERF_{ARI}$ due to BC is positive in all models included

in this study, despite any differences in magnitude, with SW ERF$_{ARI}$ virtually being almost entirely responsible for the global mean total ERF$_{ARI}$ values (Tables S2 and S3). Evidently, this shows the importance of interactions between BC and incoming SW radiation to the total forcing BC induces to the Earth's climate. Total ERF$_{ALB}$ from BC is 0.00 W m$^{-2}$ on a global scale, with similar contribution from positive SW ERF$_{ALB}$ and negative LW ERF$_{ALB}$. It should be noted that this is exactly the opposite from the case in the all-aerosol, SO$_4$ and OC experiments. Models generally agree on the sign and magnitude of ERF$_{ALB}$ caused by BC with one exception: GFDL-ESM4 produces a negative total ERF$_{ALB}$ globally (Table 4) due to a stronger negative LW ERF$_{ALB}$ (Table S3). Conversely, MRI-ESM2-0 produces the strongest positive SW ERF$_{ALB}$, and ultimately controls the SW ERF$_{ALB}$ induced by anthropogenic aerosols in the model's respective piClim-aer simulation (Oshima et al., 2020).

## 3.5 Spatial distribution of the dominant ERF component

The relevant contribution of the three main ERF components (ERF$_{ARI}$, ERF$_{ACI}$ and ERF$_{ALB}$) to the total ERF was examined on a global scale and the results for the all-aerosol experiments and the individual anthropogenic aerosol species are presented in Figs. 6 and 7, respectively. The geographical distribution of the dominant SW and LW ERF component for the all-aerosol experiments are presented in Figs. S10 and S11, respectively, and for the single-aerosol-species experiments in Figs. S12 and S13, respectively, within the Supplement. The absolute values of total ERF$_{ARI}$, ERF$_{ACI}$ and ERF$_{ALB}$ were summed for every grid cell and in cases where one of these components explained at least 50% of the resulting value, while each of the other two explained less than 33% of the summation result, the corresponding grid cell was labeled after that ERF component, otherwise it was not labeled. Although this is a rather simplistic approach to examine the contribution from ARI, ACI and ALB to the total ERF a climate forcer induces, it provides some useful insight. For instance, it becomes clear that ERF$_{ACI}$ dominates over the largest part of the globe (Fig. 6), indicating that interactions between clouds and aerosols are mainly responsible for the total ERF induced by anthropogenic aerosols at TOA over a vast area extending from around 75° S to 75° N. ERF$_{ALB}$ is mainly dominant over the poles for both piClim and histSST experiments. ERF$_{ARI}$ is the largest contributor to the total ERF over the Sahel and parts of the Sahara Desert, parts of Antarctica, Greenland (mainly seen in piClim-aer) and the Arabian Desert (in histSST). However, there are regions over the Sahara and Arabian Deserts, and Antarctica that do not exhibit a clear dominance of a single ERF component, suggesting that various processes influence the overall radiative forcing and should be attributed to more than one ERF component.

In the piClim-SO$_2$ simulation (Fig. 7a), even though ERF$_{ACI}$ dominates globally, there is a larger contribution from ERF$_{ALB}$ to the total ERF over the Arctic, the Sahara Desert and Antarctica than in piClim-aer. Moreover, ERF$_{ARI}$ loses its dominant role over Greenland and is sparsely scattered over the Sahel, the southern parts of the North Atlantic and the northwestern part of the Indian Peninsula. There is a wide region extending from the tropical North Atlantic to South Asia where more than one ERF component contributes significantly to the total ERF (Fig. 7a). OC ERF$_{ALB}$ has a more (less) pronounced dominance over Antarctica (the Arctic), along with larger contribution to the total OC ERF over continental Asia and parts of Africa (Fig. 7b) than in piClim-SO$_2$. OC ERF$_{ARI}$ is dominant over different regions than in piClim-SO$_2$ and piClim-aer, as it explains more than half of the total ERF over central South America, the Maritime Continent and areas surrounding Northern India.

In contrast with the results above, ERF$_{ARI}$ is the dominant contributor to the total ERF induced by BC over extended continental areas around the globe and the western North Pacific Ocean (Fig. 7c). While BC ERF$_{ALB}$ dominates over a large part of Antarctica, and the western and eastern parts of South Indian Ocean, BC ERF$_{ARI}$ controls the total BC ERF over the largest part of the Arctic. BC ERF$_{ARI}$ dominance is prominent over emission regions of Eastern U.S., Eastern Europe, and East and South Asia, as well as the Arabian Desert and most parts of Africa. However, in many parts of Eurasia and the Pacific Ocean the total BC ERF cannot be explained by a single ERF component. Interestingly enough, BC ERF$_{ACI}$ dominance is confined over oceanic regions for the most part (Fig. 7c).

## 3.6 AOD and ERF changes throughout the historical period

In the previous sections, only the global mean ERFs for 1995-2014 have been presented. However, it is important to examine the magnitude of transient ERF induced by anthropogenic aerosols over the entire historical period (1850-2014) for assessing the evolving aerosol radiative forcing on global and regional scale. To this end, the method proposed by Ghan (2013) was used to decompose the ERF caused by anthropogenic aerosols over the historical period. Along with the global mean ERF, five regions of interest were chosen from the IPCC AR6 WGI ATLAS (Gutiérrez et al., 2021) for investigation, namely East North America (ENA), West and Central Europe (WCE), the Mediterranean (MED), East Asia (EAS) and South Asia (SAS). The boundaries of each region are shown in the embedded maps within Figs. 8a and 9a.

The differences in pre-industrial to present-day ambient AOD at 550 nm (Fig. 8) have an increasing trend since the 1900s on global scale, but with a much smaller rate since the 1990s (Fig. 8a). Sulphate AOD has undergone the largest increase

since the pre-industrial era, followed by organic aerosol AOD on a global scale and over all the five ATLAS regions. Changes in AOD over ENA, WCE, and MED reached their peak around the late 1970s – early 1980s, with declining trends afterwards (Fig. 8b-d). On the other hand, AOD changes over EAS and SAS have been following an upward trend since the 1950s (Fig. 8e, f). Although trends from CMIP6 models after around 2010 are more difficult to assess (as historical simulations end at 2014), the decrease in anthropogenic $SO_2$ emissions over EAS since 2011 was underestimated in the CMIP6 emissions database available at the time of the CMIP6 aerosol simulations (Hoesly et al., 2018), implying that the AOD changes over EAS may not be captured precisely by CMIP6 models (Wang et al., 2021). There is a robust signal for declining anthropogenic aerosol emissions since 2000, particularly over North America, Europe, and East Asia (Quaas et al., 2022). The global and regional mean values of ΔAOD for each model can be found in Table S5. Moreover, the inter-model variability (one standard deviation) of the multi-model ensemble ΔAOD is shown in Table S5 and Fig. S14. During 1965-1984 (negative peak period; hereafter denoted as NPP), Central and Eastern Europe exhibit the largest standard deviation of ΔAOD (Fig. S14), whereas during EHP EAS shows the largest variability (Table S5).

Changes in AOD can be linked to changes in aerosol abundances and/or emissions, which in turn induce radiative forcings at TOA. This can be supported by the temporal evolution of total ERF and its components throughout the historical period (Fig. 9, and Figs. S15 and S16 in the Supplement). Globally, anthropogenic aerosol ERF attains its most negative values around the late-1980s, with a trend towards less negative values by the end of the historical period (Fig. 9a) due to regulations and restrictions in aerosol and aerosol precursor emissions (Myhre et al., 2017; Szopa et al., 2021). The dominant role of $ERF_{ACI}$ is obvious here as it closely follows total ERF, whereas $ERF_{ARI}$ and $ERF_{ALB}$ show much smaller changes. The global mean total ERF slightly decreases from -1.27 W m$^{-2}$ during NPP to -1.28 W m$^{-2}$ during EHP. There is a disagreement between models in the sign of ERF change from NPP to EHP (Table 5) as half the models (CNRM-ESM2-1, GFDL-ESM4 and NorESM2-LM) show an increase in ERF magnitude during EHP. This difference between the regional findings of IPCC AR6 of the WGI (Szopa et al., 2021) and this study can be attributed to the differences in climate models used in this ensemble (Table 2), and temporal windowing effects. $ERF_{ARI}$ becomes less negative from NPP to EHP (-0.13 W m$^{-2}$ to -0.08 W m$^{-2}$); this is a robust change among all models used here (Table 5). However, $ERF_{ACI}$ becomes more negative through time (from -1.17 W m$^{-2}$ in NPP to -1.24 W m$^{-2}$ in EHP), while $ERF_{ALB}$ gets more positive (from 0.03 W m$^{-2}$ in NPP to 0.04 W m$^{-2}$ in EHP), with most models agreeing on the sign of change. If a narrower time period was chosen (e.g., 2005-2014), the decrease in ERF magnitude towards the end of the historical period would be much more prominent (Table S6).

During the late-1970s and early-1980s total ERF and $ERF_{ACI}$ reach a negative peak over ENA, WCE and MED regions, with a simultaneous change in $ERF_{ARI}$ towards more negative values (Fig. 9b-d). Each of the three regions shows a substantial change in total ERF from NPP to EHP (an increase by +2.20 W m$^{-2}$ for ENA, +4.10 W m$^{-2}$ for WCE and +1.41 W m$^{-2}$ for MED; Table 5), along with a change towards more positive (negative) values for $ERF_{ARI}$ and $ERF_{ACI}$ ($ERF_{ALB}$). EAS exhibits a strongly decreasing trend in total ERF (Fig. 9e), with the magnitude of $ERF_{ACI}$ being extremely close to, but slightly more negative than the total ERF, while $ERF_{ARI}$ and $ERF_{ALB}$ remain almost unchanged. Total ERF becomes more negative towards the end of the historical period over EAS (from -4.28 W m$^{-2}$ in NPP to -6.36 W m$^{-2}$ in EHP) largely due to $ERF_{ACI}$ changes (an increase from -4.17 W m$^{-2}$ in NPP to -6.05 W m$^{-2}$ in EHP). Finally, over the SAS region there is a negative, ever-growing in magnitude total ERF and $ERF_{ACI}$ since the 1960s, while there is a pronounced increasing (decreasing) trend in $ERF_{ALB}$ ($ERF_{ARI}$) from the late 1980s onwards (Fig. 9f). Regional mean of total ERF, $ERF_{ARI}$, $ERF_{ACI}$, and $ERF_{ALB}$ change by -1.47 W m$^{-2}$, -0.84 W m$^{-2}$, -1.75 W m$^{-2}$, and 1.13 W m$^{-2}$, respectively, from NPP to EHP over the SAS region. Note that not all models used in this work agree on the magnitude and/or the sign of the changes described above, as some of them may under- or overestimate the influence certain physical processes exert on radiative forcings at TOA (Table 5).

Analysis of the inter-model variability (one standard deviation) of ERF over a number of IPCC AR6 WGI ATLAS regions defined in Gutiérrez et al. (2021) shows that $ERF_{ACI}$ due to all anthropogenic aerosols is the main source of uncertainty of total ERF (Table 5, Fig. S17). During EHP, the standard deviation of total $ERF_{ACI}$ is estimated at 0.44 W m$^{-2}$ globally, whereas the standard deviations of total $ERF_{ARI}$ and total $ERF_{ALB}$ are 0.14 W m$^{-2}$ and 0.08 W m$^{-2}$, respectively (Table 5). EAS contributes most to the inter-model spread of both $ERF_{ARI}$ and $ERF_{ACI}$ with an area-weighted mean standard deviation of 1.03 W m$^{-2}$ and 3.71 W m$^{-2}$, respectively (Table 5), followed by SAS, which has a much smaller standard deviation (0.86 W m$^{-2}$ and 1.76 W m$^{-2}$, respectively). Based on Fig. S17, the area-weighted standard deviation was also calculated as a percent. EAS contributes 9.2% and 6.5% to the standard deviation of total $ERF_{ARI}$ and total $ERF_{ACI}$, respectively, making it the land region with the largest contribution to both $ERF_{ARI}$ and $ERF_{ACI}$ (and total ERF as a result) standard deviations by far. The inter-model variability of total ERF (Fig. S17) mainly stems from the larger standard deviation of SW ERF (Fig. S18) rather than LW ERF (Fig. S19), with SW $ERF_{ACI}$ being the main contributor. Total ERF and total $ERF_{ACI}$ exhibit a small standard deviation during EHP over remote oceanic regions (with low ΔAOD), such as the Arctic Ocean (0.96 W m$^{-2}$ and 0.60 W m$^{-2}$, respectively), the Western South Pacific Ocean (0.23 W m$^{-2}$ and 0.37 W m$^{-2}$, respectively), the Central and Eastern South Pacific Ocean (0.35 W m$^{-2}$ and 0.34 W m$^{-2}$, respectively), the South Atlantic Ocean (0.52 W m$^{-2}$ and 0.44 W m$^{-2}$, respectively), the South Indian Ocean (0.59 W m$^{-2}$ and 0.65 W m$^{-2}$, respectively), and the Southern Ocean (0.29 W m$^{-2}$ and 0.28 W m$^{-2}$, respectively). Oceanic regions in the outflow (with high to medium ΔAOD) show a larger inter-model spread in total ERF and total $ERF_{ACI}$, such as

the Western North Pacific Ocean (1.55 W m$^{-2}$ and 1.66 W m$^{-2}$, respectively), the Central and Eastern North Pacific Ocean (1.24 W m$^{-2}$ and 1.32 W m$^{-2}$, respectively), the North Atlantic Ocean (1.14 W m$^{-2}$ and 1.28 W m$^{-2}$, respectively), the Arabian Sea (1.19 W m$^{-2}$ and 2.04 W m$^{-2}$, respectively), and the Bay of Bengal (1.73 W m$^{-2}$ and 1.72 W m$^{-2}$, respectively). There are some oceanic regions that largely contribute to the area-weighted standard deviations of total ERF and total ERF$_{ACI}$, such as the Central and Eastern North Pacific Ocean (8.5% and 8.8%, respectively), the Southern Ocean (6.8% and 6.6%, respectively), the Central and Eastern South Pacific Ocean (6.0% and 6.0%, respectively), and the North Atlantic Ocean (5.9% and 6.4%, respectively), but this is due to their large surface areas affecting their contribution percentage. Regions with large standard deviation in total ERF and ERF$_{ACI}$ over land can also be found, like N.W. South America (2.21 W m$^{-2}$ and 2.17 W m$^{-2}$, respectively), the Tibetan Plateau (1.06 W m$^{-2}$ and 1.57 W m$^{-2}$, respectively), N. South America (1.38 W m$^{-2}$ and 1.61 W m$^{-2}$, respectively), Central South America (1.79 W m$^{-2}$ and 1.60 W m$^{-2}$, respectively), and East Europe (1.07 W m$^{-2}$ and 1.36 W m$^{-2}$, respectively). It is interesting to note that the Arabian Peninsula shows a large inter-model variability in total ERF (1.33 W m$^{-2}$) during EHP, which originates from a large inter-model spread in ERF$_{ARI}$ (0.81 W m$^{-2}$) and ERF$_{ALB}$ (0.79 W m$^{-2}$) rather than ERF$_{ACI}$ (0.46 W m$^{-2}$). The land regions that exhibit the smallest standard deviation in both total ERF and total ERF$_{ACI}$ are West Antarctica (0.19 W m$^{-2}$ and 0.22 W m$^{-2}$, respectively) and East Antarctica (0.36 W m$^{-2}$ and 0.06 W m$^{-2}$, respectively). Generally, regions with high to medium $\Delta$AOD over land (Fig. 1) tend to show larger inter-model variability in total ERF and total ERF$_{ACI}$ (Fig. S17) than land regions with medium to low $\Delta$AOD, with the lowest inter-model spread appearing over remote oceanic regions with medium to low $\Delta$AOD.

Figures 10 and 11 show the regional SW and LW ERF decomposition for the five ATLAS regions presented above over the NPP (1965-1984) and EHP (1995-2014), respectively. These figures are a variation of Fig. 6.10 of IPCC AR6 WGI Chapter 6 (Szopa et al., 2021), which had a longitude mapping error in its figure plotting code (IPCC AR6 WGI Errata: https://www.ipcc.ch/report/ar6/wg1/downloads/report/IPCC_AR6_WGI_Errata.pdf). Figures 10 and 11 summarize succinctly the findings described earlier, that over EAS, and SAS, the total ERF becomes more negative in the EHP compared to the NPP, with the highest contributor from ERF$_{ACI}$, and attributed to increasing AOD towards the EHP. Over ENA, WCE and MED, the ERF becomes less negative from NPP to EHP, as observed in Fig. 9. Interestingly, over EAS and SAS, the LW ERF$_{ACI}$ is negative, while for ENA, WCE, and MED, the LW ERF$_{ACI}$ is positive. This effect is not dependent on the time period and there is no significant amplitude change in EAS and SAS LW ERF$_{ACI}$ between NPP (Fig. 10) and EHP (Fig. 11). Positive LW ERF$_{ACI}$ could be attributed to increased cloud cover with droplet sizes more likely to absorb infrared or scatter LW back towards the surface (Kuo et al., 2017). Considering that relatively higher clouds can trap outgoing LW radiation, thus leading to a positive LW ERF (and warming) it would be expected to have more higher clouds over MED and ENA and less higher clouds over EAS and SAS. Investigation of the ice water path (IWP; Fig. S20) shows that there is a decrease over EAS and SAS (i.e., less high clouds), an increase over MED and ENA (i.e., more high clouds), and a near-zero change in IWP over WCE. Liquid water path (LWP; Fig. S20) increases over EAS and SAS during EHP, while it decreases over ENA, WCE and MED during the same period. The same happens for SW ERF$_{ACI}$, which is more negative (positive) over EAS and SAS (ENA, WCE and MED) during EHP (Fig. 11). These model variables (IWP and LWP) are only indicators of the ERF changes over time and cannot fully explain the ERF time evolution during the end of the historical period. As a caveat, Burrows et al. (2022) express low confidence in global climate models' skill in simulating cloud processes, including aerosol chemistry and physics interactions.

# 4. Conclusions

The global spatial patterns of present-day effective radiative forcing (ERF) due to anthropogenic aerosols were investigated using prescribed-SST simulations from seven different ESMs participating in the CMIP6, based on both time-slice pre-industrial perturbation experiments (piClim) and transient simulations over the historical period (histSST). Shortwave (SW), longwave (LW) and total (i.e., SW plus LW) ERF and changes in aerosol optical depth (AOD) were quantified for all anthropogenic aerosols, combined and individually, using both piClim and histSST experiments for comparison purposes. Additionally, the robustness of the multi-model ensemble results was calculated by investigating both the statistical significance of each model's results and the agreement between individual models on the sign of change. Spatial patterns and temporal evolution of ERF and $\Delta$AOD were presented on global and regional scale, along with tables that show the area-weighted mean values and standard deviation of ERF and $\Delta$AOD for the multi-model ensemble as well as every individual model.

Global AOD has increased since 1850, especially over the industrialized regions of NH, reflecting the increase in anthropogenic aerosol (and precursor) emissions since the pre-industrial era (Gulev et al., 2021; Szopa et al., 2021). The highest increase in AOD was found for sulphates, followed by organic carbon (OC) and black carbon (BC) aerosols, mainly over East and South Asia.

The total ERF due to present-day anthropogenic aerosols was calculated at -1.11 ± 0.26 (one standard deviation; inter-model variability) W m$^{-2}$ using the piClim-aer experiment. It is globally negative, with more negative values over the Northern than the Southern Hemisphere. Pronounced negative ERF peaks were observed mainly over regions with aerosol emission sources and downwind, whereas ERF attains positive values over reflective surfaces. The calculated values for ERF$_{ARI}$, ERF$_{ACI}$, and ERF$_{ALB}$ are -0.02 ± 0.20 W m$^{-2}$, -1.14 ± 0.33 W m$^{-2}$, and 0.05 ± 0.07 W m$^{-2}$, respectively, with ERF$_{ACI}$ dominating the
spatial pattern of the total ERF at TOA. Other multi-model studies that used piClim experiments (e.g., Smith et al., 2020; Zanis et al., 2020; Thornhill et al., 2021) produced similar results, despite any differences in the climate model ensembles or calculation method. ERF estimates from single-model studies (e.g., Horowitz et al., 2020; Michou et al., 2020; Oshima et al., 2020; O'Connor et al., 2021) may vary from other multi-model ensemble studies because each climate model treats aerosol and cloud processes differently, and as a result they may overestimate or underestimate ARI and/or ACI (Bellouin et al., 2020;
Forster et al., 2021).
       Based on the histSST experiments, the global mean historical aerosol ERF was estimated at -1.28 ± 0.37 W m$^{-2}$ for 1995-2014 relative to pre-industrial, showing a slight, but statistically insignificant, increase in magnitude compared to the 1965-1984 mean value of -1.27 ± 0.43 W m$^{-2}$. These estimates are in good agreement with the IPCC AR6 WGI ERF assessment of -1.3 (-2.0 to -0.6) W m$^{-2}$ for 1750-2014 using multiple lines of evidence (Forster et al., 2021), but show a slight disagreement
in the sign of ERF change due to different climate models participating in this study. The estimated values of ERF$_{ARI}$, ERF$_{ACI}$, and ERF$_{ALB}$, averaged over the 1995-2014 period, are -0.08 ± 0.14 W m$^{-2}$, -1.24 ± 0.44 W m$^{-2}$, and 0.04 ± 0.08 W m$^{-2}$, respectively. The piClim-aer and the histSST experiments show remarkable similarities in their calculated global mean ERF values (total ERF and its components) and their global spatial patterns. The impact of aerosol-cloud interactions on the total ERF is highlighted, as peak negative ERF$_{ACI}$ regions coincide with the ones of total ERF for both types of experiments.
The global spatial patterns of total ERF and its components from individual aerosol species, such as sulphates, organic carbon (OC), and black carbon (BC), were also calculated based on piClim experiments. Sulphates exert a negative ERF globally (-1.11 ± 0.31 W m$^{-2}$) driving the spatial distribution of the anthropogenic aerosol forcing at TOA. It is mostly negative over emission sources of the NH, predominantly over East and South Asia. ERF$_{ACI}$ is the dominant SO$_4$ ERF component (-0.83 ± 0.23 W m$^{-2}$), and peaks over East Asia, with significant contributions from a negative ERF$_{ARI}$ (-0.32 ± 0.12 W m$^{-2}$)
particularly over South and East Asia. The total ERF due to OC is also negative, although much weaker in magnitude (-0.35 ± 0.21 W m$^{-2}$) than the ERF of sulphates, becoming more negative over East Asia and Indonesia. Conversely, BC causes a globally positive ERF (0.19 ± 0.18 W m$^{-2}$) owing to a quite strong ERF$_{ARI}$ (0.39 ± 0.19 W m$^{-2}$) all over the globe, especially over East Asia, followed by South Asia. The global estimates of ERF values are in line with those of Thornhill et al. (2021) for the same experiments.
In the all-aerosol, SO$_2$ and OC experiments, the negative SW component is responsible for the resulting total ERF$_{ARI}$ and ERF$_{ACI}$ values, as it is larger in magnitude than its positive LW counterpart, whereas the opposite is true for the total ERF$_{ALB}$ values. In the case of BC, both the SW and the LW ERF$_{ARI}$ values are positive, while the combination of a weaker, negative SW ERF$_{ACI}$ and a stronger, positive LW ERF$_{ACI}$ leads to a small, globally negative total ERF$_{ACI}$. The total ERF$_{ALB}$ is positive (as in the other experiments), because of the positive SW ERF$_{ALB}$, which is stronger than its negative LW counterpart.
It should be highlighted that the above results vary among the individual ESMs (see also Michou et al., 2020; Oshima et al., 2020; O'Connor et al., 2021; Thornhill et al., 2021).
       To determine the processes contributing the most to the total ERF on a global scale, a novel method was followed, in which each of the three main ERF components was tested whether it could explain at least half of the total ERF value. When considering all anthropogenic aerosols, ACI dominates over the largest part of the globe. ALB is most significant mainly over
the poles, while ARI prevails over certain reflective surfaces. For sulphates and OC aerosols ACI dominates, but in piClim-BC ARI dominates over the majority of NH, and especially the Arctic, while ACI clearly dominates over oceanic areas.
       Changes in AOD and ERF magnitude were investigated globally and over five NH regions of interest throughout the historical period (1850-2014). AOD shows a decreasing trend after around 1980 over East North America, West and Central Europe, and the Mediterranean (see also Bauer et al., 2020; Cherian and Quaas, 2020; Gulev et al., 2021), with a subsequent
increasing trend of anthropogenic aerosol ERF towards more positive values over those regions (see also Lund et al., 2018a; Seo et al., 2020; Smith et al., 2021; Szopa et al., 2021; Quaas et al., 2022) due to changes in anthropogenic aerosol emissions (Myhre et al., 2017). On the contrary, AOD shows a continuous increase over South Asia and East Asia after the 1950s, along with a strengthening of the total ERF. However, it is argued that CMIP6 models fail to capture the observed AOD trends over Asia towards the end of the historical simulations (Li et al., 2017; Zheng et al., 2018; Wang et al., 2021) due to inaccuracies
in the Community Emissions Data System (CEDS; Hoesly et al., 2018), which is used by many CMIP6 climate models.
       Finally, the inter-model variability of ERF and its main components (ARI, ACI, and ALB) was investigated over a number of oceanic and land regions. Our analysis indicates that ERF$_{ACI}$ is the main source of uncertainty in total ERF. More specifically, the large standard deviation of SW ERF (mainly SW ERF$_{ACI}$) dominates the spatial pattern of the inter-model spread of total ERF, with small contributions from LW ERF. East Asia is the greatest contributor to the inter-model variability
of both ERF$_{ARI}$ and ERF$_{ACI}$, while other regions, such as N.W. South America, the Arabian Sea, South Asia, and the Bay of

Bengal significantly contribute to the large standard deviation of ERF$_{ACI}$. Oceanic regions with medium to low ΔAOD show the smallest standard deviation in both total ERF and total ERF$_{ACI}$, whereas land regions with high to medium ΔAOD generally exhibit larger inter-model variability.

Overall, our results highlight the dominant role of sulphates on the ERF of anthropogenic aerosols. ERF follows the changes of aerosols from the preindustrial era onwards, exhibiting different trends over different regions around the globe. ERF$_{ACI}$ clearly dominates over ERF$_{ARI}$ and ERF$_{ALB}$ driving the ERF patterns and trends. This finding, in line with the latest IPCC WGI assessment report (AR6) (Forster et al., 2021; Szopa et al., 2021) constitutes a major update with respect to AR5 where ERF$_{ARI}$ and ERF$_{ACI}$ were considered of the same magnitude on a global scale (Boucher et al., 2013; Myhre et al., 2013).

# Appendix A

In this section, the CMIP6 variables used in this study for the ERF decomposition and the calculation of AOD changes are presented in Table A1. All data were downloaded from the ESGF node (https://esgf-node.llnl.gov/search/cmip6/, last access: August 31$^{st}$, 2023). Moreover, the method of determining the robustness of the ΔAOD and the ERF results presented in Figures 1-3 is described in Table A2.

*Data availability*. All data from the Earth System Models used in this paper are available on the Earth System Grid Federation website and can be downloaded from there (https://esgf-node.llnl.gov/search/cmip6/, ESGF, 2023).

*Author contributions*. AK and PZ conceptualized this study. AK, PZ, AKG and DA designed the analysis. AK performed the formal analysis, produced the figures (Figs. 10 and 11 were produced by CK) and prepared the original draft. PN contributed to CNRM-ESM2-1 simulations; TvN and PLS contributed to EC-Earth3-AerChem simulations; LWH and VN contributed to GFDL-ESM4 simulations; DN contributed to MPI-ESM-1-2-HAM simulations; NO contributed to MRI-ESM2-0 simulations; DO contributed to NorESM2-LM simulations; JM contributed to UKESM1-0-LL simulations. All authors contributed to the revision and editing of the paper.

*Competing interests*. The authors declare that they have no conflict of interest.

*Acknowledgements*. The authors acknowledge the World Climate Research Program, which promoted and coordinated CMIP6 through its Working Group on Coupled Modelling. The authors thank the climate modelling groups (listed in Table 1) for producing and making available their model output, the Earth System Grid Federation (ESGF) for archiving the data and providing access, and the multiple funding agencies who support CMIP6 and ESGF. The authors acknowledge input from Paul Glantz for improvements in the methodology for creating Figure 6.10b in IPCC WGI AR6 Chapter 6 which then lead to Figures 10 and 11 with the improved methodology.

*Financial support*. AK, DA, and PZ acknowledge support from the REINFORCE (improvements in the simulation of aerosol-climate linkages in earth system models: from global to Regional scalEs) research project; the research project is implemented in the framework of H.F.R.I call "Basic research Financing (Horizontal support of all Sciences)" under the National Recovery and Resilience Plan "Greece 2.0" funded by the European Union – NextGenerationEU (H.F.R.I. Project Number: 15155). NO was supported by the Environment Research and Technology Development Fund (JPMEERF20202003 and JPMEERF20232001) of the Environmental Restoration and Conservation Agency provided by Ministry of the Environment of Japan, the Arctic Challenge for Sustainability II (ArCS II), Program Grant Number JPMXD1420318865, and a grant for the Global Environmental Research Coordination System from Ministry of the Environment, Japan (MLIT2253). TvN, PLS and DN acknowledge funding from the European Union's Horizon 2020 research and innovation program under grant agreement No 821205 (FORCeS).

*Review statement*. This paper was edited by Guy Dagan and reviewed by two anonymous referees.

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

**Table 1.** Estimates of present-day aerosol Effective Radiative Forcing (in W m⁻²) from recent papers. ERF values were calculated for all aerosols, sulphates, black carbon, and organic aerosols (organic carbon, primary organic matter, secondary organic aerosols).

| Paper | Model(s) | Method | Period | Aerosols | | | | Sulphates | | | | Black Carbon | | | | Organic Aerosols / Organic Carbon | | | |
|---|---|---|---|---|---|---|---|---|---|---|---|---|---|---|---|---|---|---|---|
| | | | | ERF | ACI | ARI | ALB | ERF | ACI | ARI | ALB | ERF | ACI | ARI | ALB | ERF | ACI | ARI | ALB |
| Albright et al. (2021) | 10 CMIP6 models | Stevens (2015) | 2000-2010 relative to 1750 | −0.95 (−1.40 to −0.56) | − | − | − | − | − | − | − | − | − | − | − | − | − | − | − |
| | | | 2010-2019 relative to 1750 | −0.85 (−1.30 to −0.50) | − | − | − | − | − | − | − | − | − | − | − | − | − | − | − |
| Bellouin et al. (2020) | Multiple lines of evidence | − | Present-day relative to 1850 | −3.15 to −0.35 (−2.00 to −0.35) | −2.65 to −0.007 | −0.71 to −0.14 | − | − | − | − | − | − | − | − | − | − | − | − | − |
| Fiedler et al. (2023) | 21 CMIP6 models | Forster et al. (2016) | 2014 relative to 1850 | −1.06 | − | − | − | − | − | − | − | − | − | − | − | − | − | − | − |
| | 12 CMIP6 models | Ghan (2013) | 1850 | −1.08* | −1.12* | 0.00* | 0.04* | − | − | − | − | − | − | − | − | − | − | − | − |
| Michou et al. (2020) | CNRM-CM6-1 | Ghan (2013) | 2014 relative to 1850 | −1.16 | −0.79 | −0.42 | 0.06 | −0.75 | −0.53 | −0.29 | 0.08 | 0.11 | − | 0.13 | 0.01 | −0.07 | − | −0.14 | 0.04 |
| | CNRM-ESM2-1 | Ghan (2013) | 1850 | −0.74 | −0.61 | −0.21 | 0.08 | − | − | − | − | − | − | − | − | − | − | − | − |
| O'Connor et al. (2021) | UKESM1 | Ghan (2013) | 2014 relative to 1850 | −1.09 ± 0.04 | −1.00 ± 0.02 | −0.10 ± 0.02 | − | −1.37 ± 0.03 | −0.91 ± 0.02 | −0.46 ± 0.02 | 0.37 ± 0.03 | 0.24 | −0.01 ± 0.02 | 0.38 ± 0.02 | − | −0.22 ± 0.04 | −0.07 ± 0.02 | −0.14 ± 0.03 | − |
| Oshima et al. (2020) | MRI-ESM2.0 | Ghan (2013) | 2014 relative to 1850 | −1.22 | −0.98 | −0.32 | 0.08 | −1.38 | −0.94 | −0.48 | 0.05 | 0.25 | −0.09 | 0.25 | 0.07 | −0.33 | −0.21 | −0.07 | −0.05 |
| | | APRP | 1850 | −0.76 | −0.45 | − | − | − | − | − | − | − | − | − | − | − | − | − | − |
| Seo et al. (2020) | UKESM1 | APRP | 1940-1970 relative to 1850 | −1.03 ± 0.05 | − | − | − | − | − | − | − | − | − | − | − | − | − | − | − |
| | | | 1980-2010 relative to 1850 | −0.87 ± 0.04 | − | − | − | − | − | − | − | − | − | − | − | − | − | − | − |
| Smith et al. (2020) | 17-model ensemble | APRP | 2014 relative to 1850 | −1.43 ± 0.05 | −1.17 ± 0.03 | −0.30 ± 0.01 | 0.04 ± 0.03 | − | − | − | − | − | − | − | − | − | − | − | − |
| | | Forster et al. (2016) | 1850 | −0.81 ± 0.30 | −0.23 ± 0.19 | −0.16 ± 0.01 | 0.00 ± 0.02 | − | − | − | − | − | − | − | − | − | − | − | − |
| Smith et al. (2021) | Energy balance model trained on 11 CMIP6 climate models | − | 2005-2014 relative to 1750 | −0.90 (−1.56 to −0.35) | −0.59 (−1.18 to −0.10) | −0.31 (−0.62 to −0.08) | − | − | − | − | − | − | − | − | − | − | − | − | − |
| | | | 2019 relative to 1750 | −1.10 (−1.78 to −0.48) | −0.69 (−1.36 to −0.40) | −0.40 (−0.77 to −0.12) | − | − | − | − | − | − | − | − | − | − | − | − | − |
| Thornhill et al. (2021) | 9-model ensemble | Forster et al. (2016) | 2014 relative to 1850 | −1.01 ± 0.25 | − | − | − | −1.03 ± 0.37 | − | − | − | 0.15 ± 0.17 | − | − | − | −0.25 ± 0.09 | − | − | − |
| Zanis et al. (2020) | 10-model ensemble | Forster et al. (2016) | 2014 relative to 1850 | −1.00 ± 0.24 | − | − | − | − | − | − | − | − | − | − | − | − | − | − | − |
| Zelinka et al. (2023) | 20-model ensemble | APRP | 2014 relative to 1850 | − | −0.88 ± 0.34 | −0.21 ± 0.28 | − | − | − | − | − | − | − | − | − | − | − | − | − |
| Zhang et al. (2022) | E3SM version 1 | Ghan (2013) | 2010 relative to 1850 | −1.64 | −1.77 | 0.04 | 0.09 | −1.66 | − | − | − | 0.27 | − | − | − | −0.40 (POM) −0.31 (SOA) | − | − | − |

\* As calculated from the values given in Table S3 within the Supporting Information of Fiedler et al. (2023; https://agupubs.onlinelibrary.wiley.com/action/downloadSupplement?doi=10.1029%2F2023GL104848&file=2023GL104848-sup-0001-Supporting+Information+SI-S01.pdf).

**Table 2.** Information on model resolution (horizontal and vertical), variant label and references for each ESM used in this work. Each experiment (see Table 2) has a variant label $r_a i_b p_c f_d$, where a is the realization index, b the initialization index, c the physics index and d the forcing index.

| Model | Resolution | Vertical Levels | piClim-(aer, control) variant label | piClim-(SO₂, OC, BC) variant label | histSST & histSST-piAer variant label | Indirect Effects Considered | Model References | Experiment References |
|---|---|---|---|---|---|---|---|---|
| CNRM-ESM2-1 | 1.4° x 1.4° | 91 levels, top level: 78.4 Km | r1i1p1f2 | r1i1p1f2 | r1i1p1f2* | Twomey effect only | (Séférian et al., 2019) (Michou et al., 2020) (Roehrig et al., 2020) | (Seferian, 2019b, c, d, e, f, a) |
| EC-Earth3-AerChem | 0.7° x 0.7°** | 91 levels, top level: 0.01 hPa | r1i1p1f1 | - | r1i1p1f1 | Twomey & Albrecht effects | (Döscher et al., 2022) (van Noije et al., 2021) | (Consortium (EC-Earth), 2021a, 2020a, 2021b, 2020b) |
| GFDL-ESM4 | 1.25° x 1°*** | 49 levels, top level: 0.01 hPa | r1i1p1f1 | r1i1p1f1 | r1i1p1f1 | Twomey & Albrecht effects | (Dunne et al., 2020) (Horowitz et al., 2020) | (Horowitz et al., 2018a, b, c, d, e, f, g) |
| MPI-ESM-1-2-HAM | 1.875° x 1.875° | 47 levels, top level: 0.01 hPa | r1i1p1f1 | r1i1p1f1 | r1i1p1f1 | Twomey & Albrecht effects | (Mauritsen et al., 2019) (Neubauer et al., 2019e) (Tegen et al., 2019) | (Neubauer et al., 2019a, b, 2020a, b, c, 2019c, d) |
| MRI-ESM2-0 | 1.125° x 1.125° | 80 levels, top level: 0.01 hPa | r1i1p1f1 | r1i1p1f1 | - | Twomey & Albrecht effects | (Kawai et al., 2019) (Oshima et al., 2020) (Yukimoto et al., 2019f) | (Yukimoto et al., 2019a, b, c, d, e) |
| NorESM2-LM | 2.5° x 1.875° | 32 levels, top level: 3 hPa | r1i1p2f1 | r1i1p2f1 | r1i1p2f1 | Twomey & Albrecht effects | (Kirkevåg et al., 2018) (Seland et al., 2020) | (Oliviè et al., 2019a, b, c, d, e, f, g) |
| UKESM1-0-LL | 1.875° x 1.25° | 85 levels, top level: 85 km | r1i1p1f4 | r1i1p1f4 | r1i1p1f2 | Twomey & Albrecht effects | (Archibald et al., 2020) (Mulcahy et al., 2020) (Sellar et al., 2020) (Seo et al., 2020) (Yool et al., 2020) (O'Connor et al., 2021) | (Dalvi et al., 2020a, b; O'Connor, 2019a, b, c, d, e) |

* The histSST-piAer simulation is identical to the histSST-piNTCF simulation as CNRM-ESM2-1 has no tropospheric chemistry, and therefore no ozone precursors, which means that the two configurations (histSST-piAer and histSST-piNTCF) are identical.

** The 0.7° x 0.7° is approximate for the TL255 grid of IFS. Aerosols and atmospheric chemistry are simulated with the Tracer Model version 5 (TM5) with horizontal resolution 3° x 2° (lon x lat), with 34 levels in the vertical and a model top at about 0.1
hPa.

*** GFDL-ESM4 uses a cubed-sphere grid with ~100 km resolution. Results are regridded to a 1.25° x 1° (lon x lat) grid for analysis.

**Table 3.** List of fixed-SST simulations used in this study. The histSST and histSST-piAer experiments cover the historical period (1850-2014). The piClim experiments are time-slice experiments covering 30 years in total and use pre-industrial climatological average SST and SIC. The year indicates that the emissions or concentrations are fixed to that year, while "Hist" means that the concentrations or emissions evolve as for the CMIP6 "historical" experiment (more information in Collins et al., 2017).

| Experiment | Type | CH$_4$ | N$_2$O | Aerosol precursors | Ozone precursors | CFC/HCFC | MIP |
|---|---|---|---|---|---|---|---|
| piClim-control | 30-year time-slice experiment | 1850 | 1850 | 1850 | 1850 | 1850 | RFMIP / AerChemMIP |
| piClim-aer | 30-year time-slice experiment | 1850 | 1850 | 2014 | 1850 | 1850 | RFMIP / AerChemMIP |
| piClim-BC | 30-year time-slice experiment | 1850 | 1850 | 1850 (non-BC) 2014 (BC) | 1850 | 1850 | AerChemMIP |
| piClim-OC | 30-year time-slice experiment | 1850 | 1850 | 1850 (non-OC) 2014 (OC) | 1850 | 1850 | AerChemMIP |
| piClim-SO$_2$ | 30-year time-slice experiment | 1850 | 1850 | 1850 (non-SO$_2$) 2014 (SO$_2$) | 1850 | 1850 | AerChemMIP |
| histSST | Transient simulation | Hist | Hist | Hist | Hist | Hist | AerChemMIP |
| histSST-piAer | Transient simulation | Hist | Hist | 1850 | Hist | Hist | AerChemMIP |


**Table 4.** Global mean ERF values (in W m$^{-2}$) for the piClim experiments (piClim-aer, piClim-SO$_2$, piClim-OC and piClim-BC), and the transient (histSST) experiment averaged over the 1995-2014 period. The total ERF and its decomposition into ERF$_{ARI}$, ERF$_{ACI}$ and ERF$_{ALB}$ are presented for each ESM, along with the multi-model ensemble mean and the inter-model variability (SD: one standard deviation).

| Model | piClim-aer | | | | piClim-SO$_2$ | | | | piClim-OC | | | | piClim-BC | | | | histSST (1995-2014) | | | |
|---|---|---|---|---|---|---|---|---|---|---|---|---|---|---|---|---|---|---|---|---|
| | ERF | ARI | ACI | ALB | ERF | ARI | ACI | ALB | ERF | ARI | ACI | ALB | ERF | ARI | ACI | ALB | ERF | ARI | ACI | ALB |
| CNRM-ESM2-1 | -0.74 | -0.21 | -0.61 | 0.08 | -0.74 | -0.29 | -0.53 | 0.08 | -0.17 | -0.07 | -0.14 | 0.04 | 0.11 | 0.13 | -0.03 | 0.01 | -0.86 | -0.26 | -0.59 | -0.01 |
| EC-Earth3-AerChem | -1.35 | 0.11 | -1.53 | 0.07 | - | - | - | - | - | - | - | - | - | - | - | - | -1.70 | 0.02 | -1.86 | 0.14 |
| GFDL-ESM4 | -0.70 | 0.26 | -0.92 | -0.03 | -0.67 | -0.21 | -0.51 | 0.05 | -0.21 | -0.10 | -0.16 | 0.05 | 0.35 | 0.52 | -0.09 | -0.09 | -0.79 | 0.06 | -0.87 | 0.02 |
| MPI-ESM-1-2-HAM | -1.26 | 0.16 | -1.57 | 0.14 | -1.06 | -0.24 | -0.96 | 0.14 | -0.78 | -0.02 | -0.79 | 0.02 | -0.15 | 0.72 | -0.87 | 0.00 | -1.33 | 0.08 | -1.51 | 0.11 |
| MRI-ESM2-0 | -1.23 | -0.32 | -1.00 | 0.08 | -1.39 | -0.48 | -0.96 | 0.05 | -0.34 | -0.07 | -0.22 | -0.05 | 0.23 | 0.25 | -0.10 | 0.07 | - | - | - | - |
| NorESM2-LM | -1.41 | 0.04 | -1.38 | -0.06 | -1.45 | -0.19 | -1.11 | -0.15 | -0.38 | -0.08 | -0.27 | -0.03 | 0.24 | 0.33 | -0.10 | 0.02 | -1.74 | -0.06 | -1.60 | -0.09 |
| UKESM1-0-LL | -1.10 | -0.15 | -1.00 | 0.05 | -1.36 | -0.49 | -0.90 | 0.03 | -0.21 | -0.15 | -0.06 | 0.01 | 0.37 | 0.37 | 0.00 | 0.00 | -1.28 | -0.27 | -1.10 | 0.08 |
| ENSEMBLE (Mean) | -1.11 | -0.02 | -1.14 | 0.05 | -1.11 | -0.32 | -0.83 | 0.03 | -0.35 | -0.08 | -0.27 | 0.01 | 0.19 | 0.39 | -0.20 | 0.00 | -1.28 | -0.08 | -1.24 | 0.04 |
| ENSEMBLE (SD) | 0.26 | 0.20 | 0.33 | 0.07 | 0.31 | 0.12 | 0.23 | 0.09 | 0.21 | 0.04 | 0.24 | 0.03 | 0.18 | 0.19 | 0.30 | 0.05 | 0.37 | 0.14 | 0.44 | 0.08 |


**Table 5.** Mean ERF values (in W m$^{-2}$) during the negative ERF peak period (1965-1984) and the recent past (1995-2014). Global and regional ERF estimates for the five NH regions of interest (ENA: East North America, WCE: West and Central Europe, MED: Mediterranean, EAS: East Asia, SAS: South Asia) are presented for each model, along with the multi-model ensemble mean and the inter-model variability (SD: one standard deviation).


| Model | Region | 1965-1984 | | | | 1995-2014 | | | |
|---|---|---|---|---|---|---|---|---|---|
| | | ERF | ARI | ACI | ALB | ERF | ARI | ACI | ALB |
| CNRM-ESM2-1 | ENA | -3.75 | -2.01 | -1.95 | 0.21 | -2.94 | -1.14 | -1.51 | -0.29 |
| | WCE | -3.24 | -1.95 | -1.16 | -0.13 | -1.92 | -0.63 | -1.73 | 0.44 |
| | MED | -2.74 | -1.66 | -1.61 | 0.53 | -1.64 | -0.95 | -0.78 | 0.08 |
| | EAS | -2.59 | -1.05 | -1.42 | -0.12 | -4.49 | -2.33 | -2.16 | 0.00 |
| | SAS | -1.79 | -0.84 | -0.49 | -0.46 | -3.93 | -2.61 | -1.48 | 0.15 |
| | GLOBAL | -0.68 | -0.23 | -0.49 | 0.04 | -0.86 | -0.26 | -0.59 | -0.01 |
| EC-Earth3-AerChem | ENA | -8.21 | -1.46 | -6.96 | 0.22 | -5.68 | -0.81 | -4.89 | 0.02 |
| | WCE | -8.76 | -1.48 | -7.85 | 0.57 | -2.92 | -0.55 | -2.61 | 0.24 |
| | MED | -4.44 | -1.47 | -4.19 | 1.22 | -2.17 | -0.39 | -3.05 | 1.28 |
| | EAS | -5.70 | -0.09 | -5.45 | -0.16 | -10.21 | 0.27 | -10.51 | 0.03 |
| | SAS | -3.72 | -0.10 | -3.59 | -0.03 | -4.14 | -0.30 | -6.68 | 2.84 |
| | GLOBAL | -1.93 | -0.14 | -1.81 | 0.02 | -1.70 | 0.02 | -1.86 | 0.14 |
| GFDL-ESM4 | ENA | -5.35 | -0.44 | -4.72 | -0.19 | -3.54 | -0.19 | -3.41 | 0.06 |
| | WCE | -7.16 | -0.43 | -6.50 | -0.23 | -4.42 | -0.06 | -3.82 | -0.53 |
| | MED | -3.02 | -0.78 | -2.81 | 0.57 | -2.49 | -0.32 | -2.42 | 0.25 |
| | EAS | -3.35 | 0.26 | -3.58 | -0.04 | -4.42 | 0.84 | -5.30 | 0.03 |
| | SAS | -1.19 | 0.09 | -1.23 | -0.05 | -4.13 | -0.54 | -3.71 | 0.11 |
| | GLOBAL | -0.75 | -0.02 | -0.72 | -0.02 | -0.79 | 0.06 | -0.87 | 0.02 |
| MPI-ESM-1-2-HAM | ENA | -8.29 | -0.92 | -7.53 | 0.16 | -5.26 | -0.49 | -4.90 | 0.12 |
| | WCE | -12.63 | -1.01 | -12.55 | 0.94 | -4.88 | -0.25 | -4.90 | 0.26 |
| | MED | -5.35 | -0.75 | -5.80 | 1.19 | -3.58 | -0.21 | -4.57 | 1.20 |
| | EAS | -9.14 | 0.33 | -9.98 | 0.51 | -11.54 | 0.28 | -12.13 | 0.31 |
| | SAS | -1.90 | 0.09 | -1.57 | -0.42 | -2.80 | 0.01 | -3.46 | 0.64 |
| | GLOBAL | -1.41 | -0.03 | -1.49 | 0.11 | -1.33 | 0.08 | -1.51 | 0.11 |
| NorESM2-LM | ENA | -5.60 | -0.71 | -4.83 | -0.05 | -3.50 | -0.24 | -3.32 | 0.06 |
| | WCE | -5.96 | -0.91 | -6.10 | 1.04 | -2.51 | -0.19 | -2.89 | 0.57 |
| | MED | -2.78 | -1.20 | -2.65 | 1.07 | -2.18 | -0.44 | -1.83 | 0.08 |
| | EAS | -2.28 | -0.27 | -2.32 | 0.30 | -5.07 | -0.58 | -4.85 | 0.36 |
| | SAS | -0.99 | -0.17 | -1.67 | 0.85 | -3.12 | -0.92 | -3.34 | 1.14 |
| | GLOBAL | -1.40 | -0.08 | -1.29 | -0.03 | -1.74 | -0.06 | -1.60 | -0.09 |
| UKESM1-0-LL | ENA | -5.93 | -1.87 | -4.25 | 0.18 | -3.71 | -1.14 | -2.37 | -0.19 |
| | WCE | -4.97 | -2.11 | -2.86 | 0.00 | -2.15 | -0.70 | -0.71 | -0.73 |
| | MED | -3.56 | -2.00 | -2.15 | 0.59 | -1.97 | -1.02 | -1.95 | 1.01 |
| | EAS | -2.64 | -0.32 | -2.25 | -0.07 | -3.91 | -0.75 | -3.02 | -0.14 |
| | SAS | -1.65 | -0.14 | -1.08 | -0.43 | -1.60 | -1.38 | -1.39 | 1.17 |
| | GLOBAL | -1.45 | -0.31 | -1.19 | 0.04 | -1.28 | -0.27 | -1.10 | 0.08 |
| ENSEMBLE (Mean) | ENA | -6.19 | -1.24 | -5.04 | 0.09 | -3.99 | -0.66 | -3.29 | -0.04 |
| | WCE | -7.12 | -1.32 | -6.17 | 0.37 | -3.02 | -0.39 | -2.64 | 0.02 |
| | MED | -3.65 | -1.31 | -3.20 | 0.86 | -2.24 | -0.56 | -2.31 | 0.63 |
| | EAS | -4.28 | -0.19 | -4.17 | 0.07 | -6.36 | -0.40 | -6.05 | 0.09 |
| | SAS | -1.87 | -0.18 | -1.61 | -0.09 | -3.34 | -1.02 | -3.36 | 1.04 |
| | GLOBAL | -1.27 | -0.13 | -1.17 | 0.03 | -1.28 | -0.08 | -1.24 | 0.04 |
| ENSEMBLE (SD) | ENA | 1.61 | 0.59 | 1.84 | 0.15 | 1.00 | 0.39 | 1.23 | 0.15 |
| | WCE | 3.00 | 0.59 | 3.65 | 0.51 | 1.13 | 0.24 | 1.35 | 0.49 |
| | MED | 0.96 | 0.45 | 1.40 | 0.30 | 0.61 | 0.31 | 1.18 | 0.52 |
| | EAS | 2.45 | 0.46 | 2.90 | 0.25 | 3.06 | 1.03 | 3.71 | 0.18 |
| | SAS | 0.89 | 0.31 | 0.97 | 0.46 | 0.91 | 0.86 | 1.76 | 0.92 |
| | GLOBAL | 0.43 | 0.11 | 0.44 | 0.05 | 0.37 | 0.14 | 0.44 | 0.08 |

**Table A1.** Description of the CMIP6 variables used in this study.

| Variable | Description | Units |
|---|---|---|
| od550aer | Ambient Aerosol Optical Thickness at 550 nm | Unitless |
| od550bc | Black Carbon Optical Thickness at 550 nm | Unitless |
| od550oa | Total Organic Aerosol Optical Depth at 550 nm | Unitless |
| od550so4 | Sulphate Aerosol Optical Depth at 550 nm | Unitless |
| rlut | Top-of-Atmosphere Outgoing Longwave Radiation | $W\ m^{-2}$ |
| rlutaf | Top-of-Atmosphere Outgoing Aerosol-Free Longwave Radiation | $W\ m^{-2}$ |
| rlutcs | Top-of-Atmosphere Outgoing Clear-Sky Longwave Radiation | $W\ m^{-2}$ |
| rlutcsaf | Top-of-Atmosphere Outgoing Clear-Sky, Aerosol-Free Longwave Radiation | $W\ m^{-2}$ |
| rsut | Top-of-Atmosphere Outgoing Shortwave Radiation | $W\ m^{-2}$ |
| rsutaf | Top-of-Atmosphere Outgoing Aerosol-Free Shortwave Radiation | $W\ m^{-2}$ |
| rsutcs | Top-of-Atmosphere Outgoing Clear-Sky Shortwave Radiation | $W\ m^{-2}$ |
| rsutcsaf | Top-of-Atmosphere Outgoing Clear-Sky, Aerosol-Free Shortwave Radiation | $W\ m^{-2}$ |
| clivi | Ice Water Path | $Kg\ m^{-2}$ |
| lwp | Liquid Water Path | $Kg\ m^{-2}$ |

**Table A2.** Criteria for determining the robustness of the results presented in Figures 1-3 in the text and in Figures S1, S2, S5, S6 and S20 in the Supplement.

| Characterization | Visual implementation | Definition |
|---|---|---|
| Robust signal | Colour (no overlay) | $\geq 80\%$ of models have statistically significant results AND $\geq 80\%$ of models agree on the sign of change |
| No robust signal | Hatching ( / / ) | $< 80\%$ of models have statistically significant results AND $\geq 80\%$ of models agree on the sign of change |
| Conflicting signals | Crosses ( x x ) | $\geq 80\%$ of models have statistically significant results AND $< 80\%$ of models agree on the sign of change |
| | | $< 80\%$ of models have statistically significant results AND $< 80\%$ of models agree on the sign of change |

# Changes in AOD (ENSEMBLE)

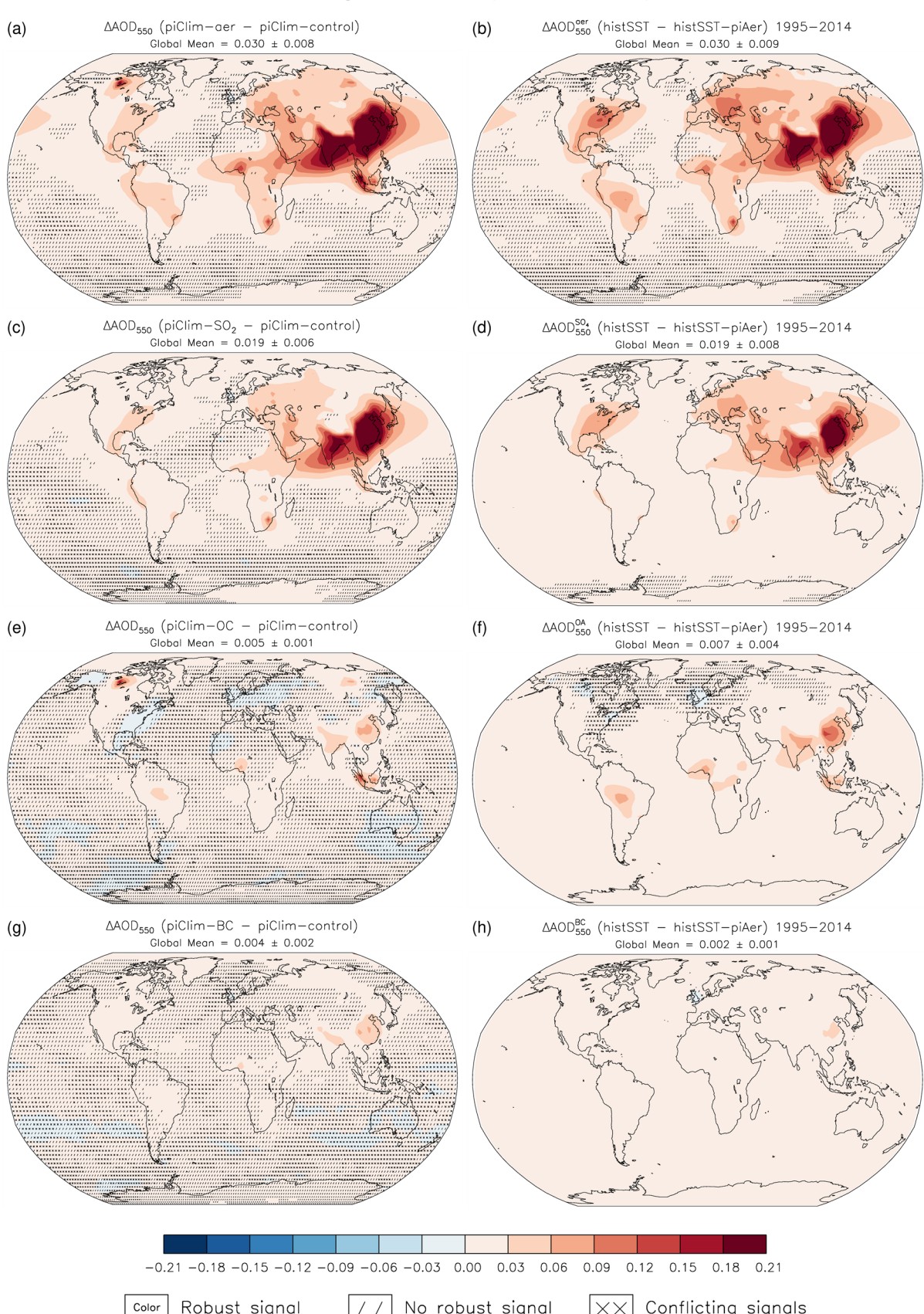

**Figure 1.** Changes in AOD at 550 nm due to all anthropogenic aerosols (1st row), SO₂ and sulphates (2nd row), OC and anthropogenic organic aerosols (3rd row), and BC (4th row) relative to the pre-industrial era. The spatial distribution is shown for the multi-model ensembles of piClim (left column) and histSST (averaged over 1995-2014; right column) experiments, respectively. The global mean ΔAOD is presented along with the inter-model variability (one standard deviation). Colored areas devoid of markings indicate robust changes, while hatched (/) and cross-hatched (X) areas indicate non-robust changes and conflicting signals, respectively. In subplots (d), (f), and (h) only a subset of the models was analyzed (see main text).


# All Aerosols Total ERF (ENSEMBLE)

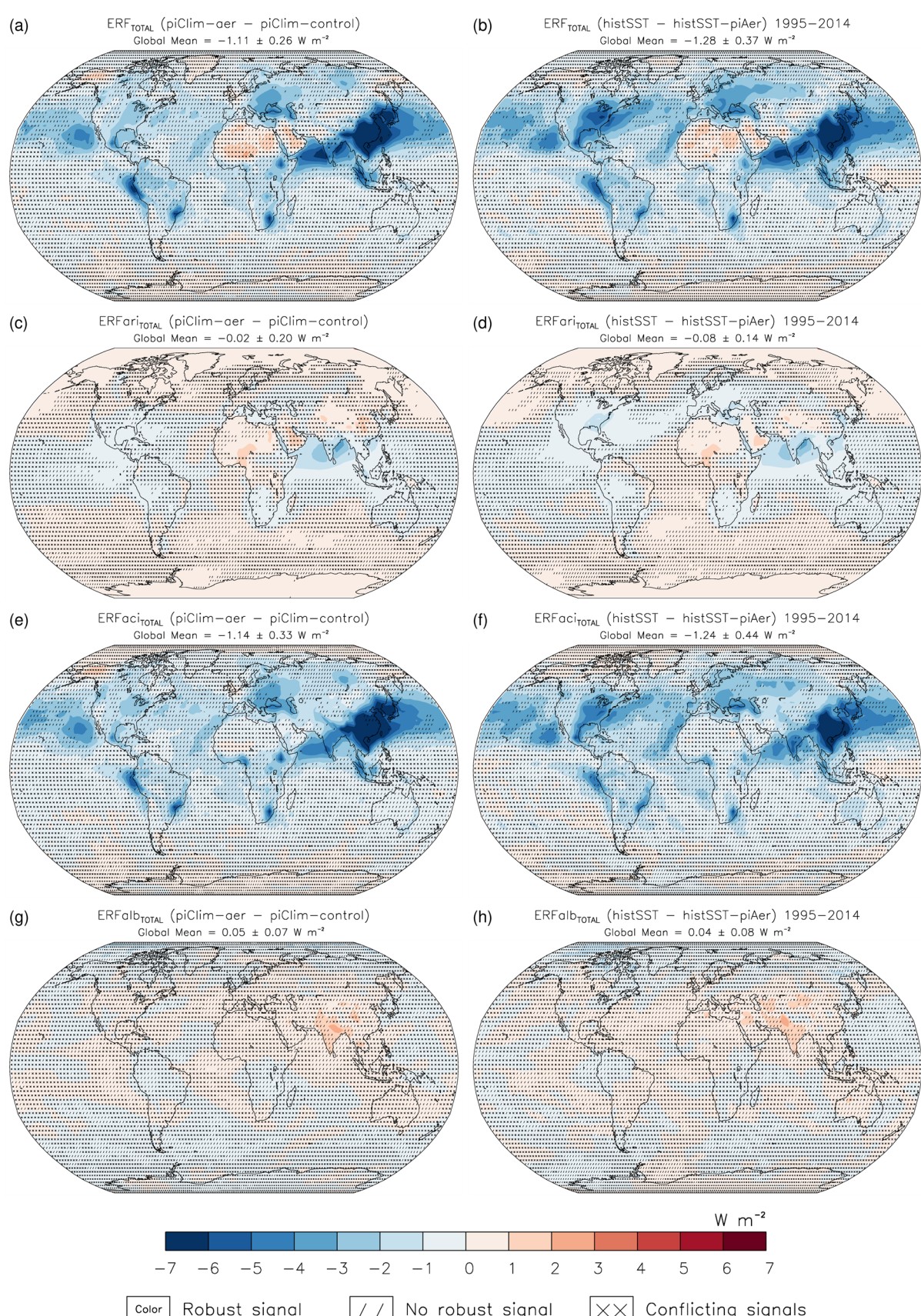

**Figure 2.** The total (SW+LW) ERF due to all anthropogenic aerosols relative to the pre-industrial era. The TOA spatial distribution is presented for the multi-model ensembles of piClim (left column) and histSST (averaged over 1995-2014; right column) experiments, respectively. The global mean total ERF (1st row), ERF$_{ARI}$ (2nd row), ERF$_{ACI}$ (3rd row), and ERF$_{ALB}$ (4th row) are shown along with the inter-model variability (one standard deviation). Colored areas devoid of markings indicate robust changes, while hatched (/) and cross-hatched (X) areas indicate non-robust changes and conflicting signals, respectively.



## Anthropogenic Aerosols Total ERF (ENSEMBLE)

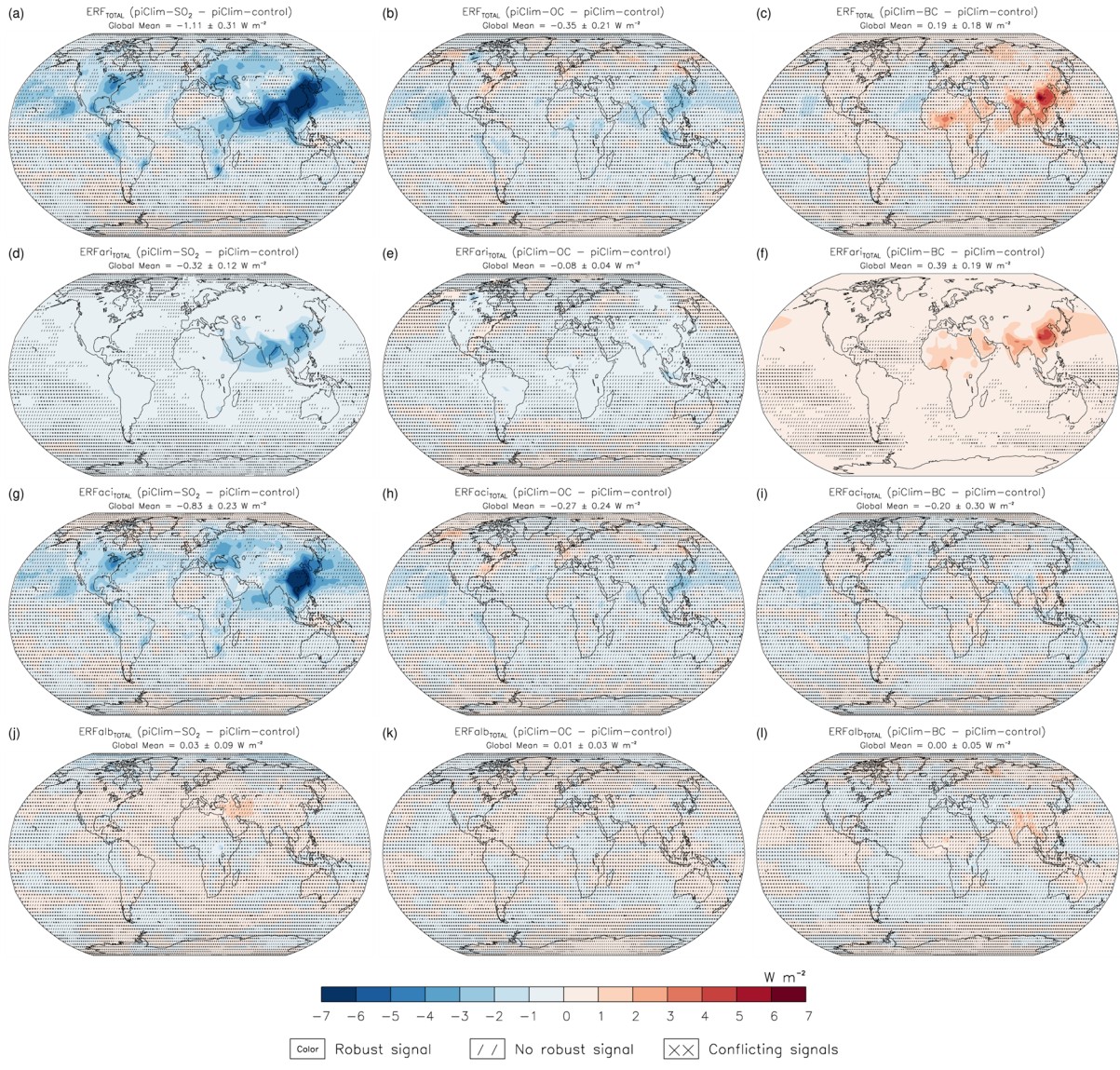

**Figure 3.** As in Fig. 2, but for piClim-SO₂ (left), piClim-OC (middle), and piClim-BC (right).

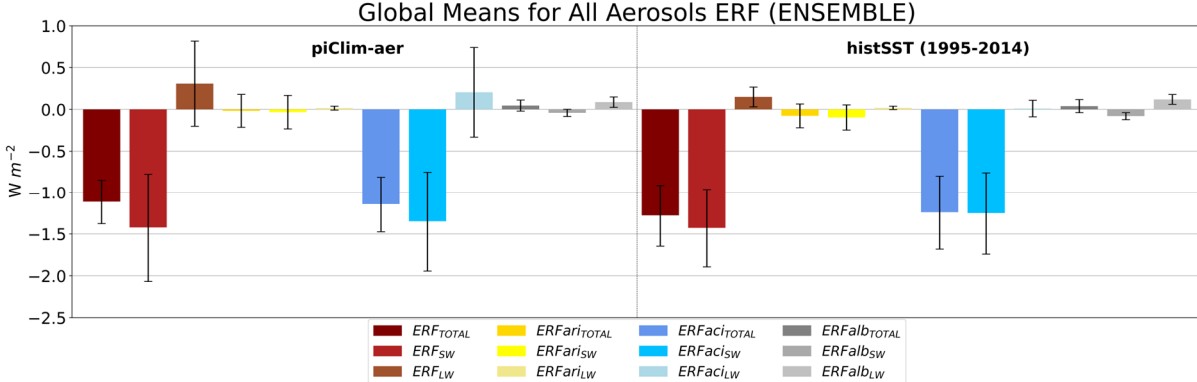

**Figure 4.** Global multi-model mean SW, LW, and total ERF values for the piClim-aer and histSST (averaged over 1995-2014) experiments. The error bars indicate inter-model variability (one standard deviation).

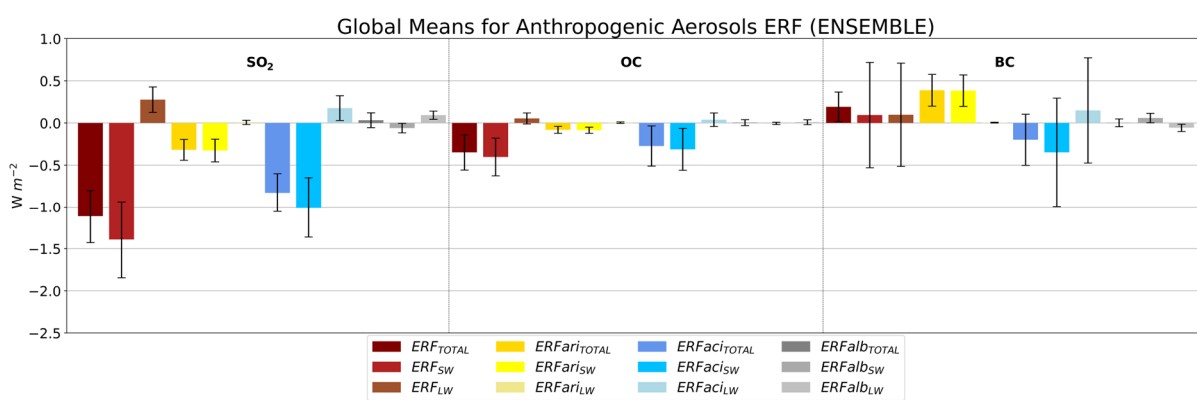

**Figure 5.** As in Fig. 4, but for piClim-SO₂, piClim-OC, and piClim-BC.

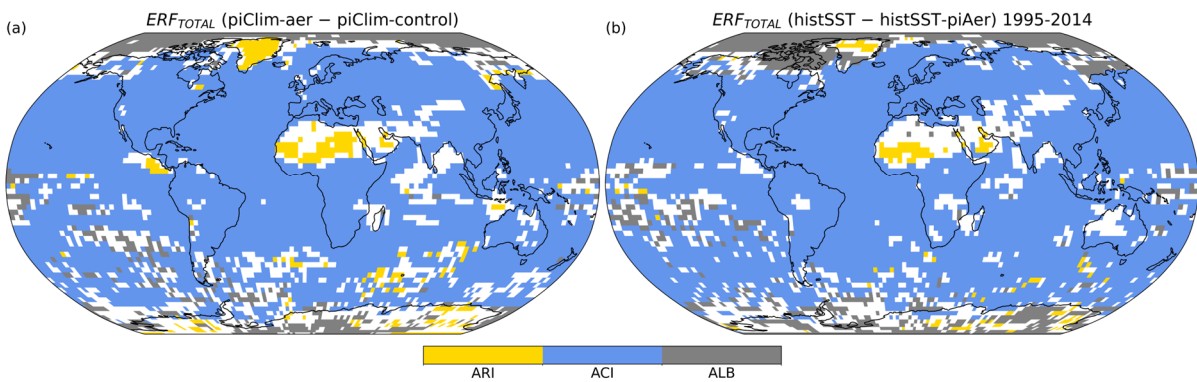

**Figure 6.** Areas where each of the three main ERF components (ERF$_{ARI}$, ERF$_{ACI}$, and ERF$_{ALB}$) dominates the total ERF. The absolute values of total ERF$_{ARI}$, total ERF$_{ACI}$, and total ERF$_{ALB}$ are summed, and every grid cell is colored after the ERF component that contributes at least 50% to the resulting value, while each of the other two components contributes less than 33% to the resulting value. In cases where the above criterion is not met, the grid cell is colored white.

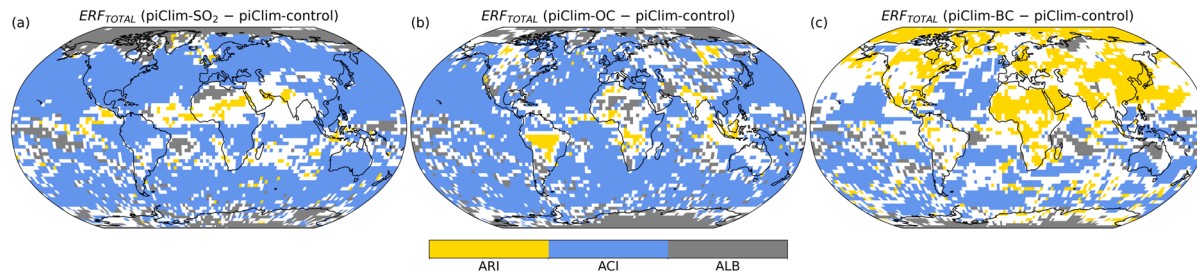

**Figure 7.** As in Fig. 6, but for piClim-SO$_2$ (left), piClim-OC (middle), and piClim-BC (right).

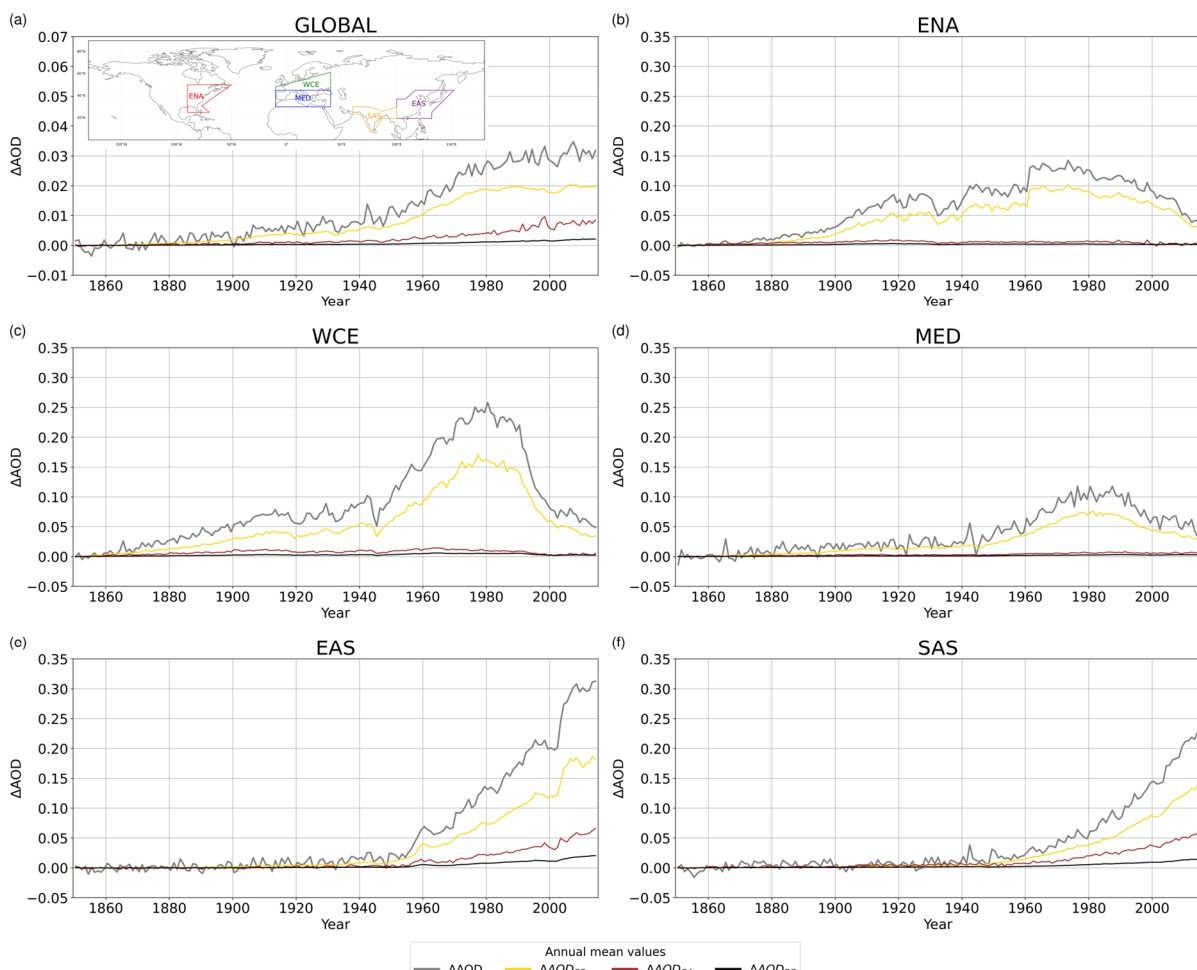

**Figure 8.** Time evolution of AOD changes due to all anthropogenic aerosols, sulphates, organic aerosols, and BC over the historical period (1850-2014). The results are presented for the histSST experiment on global scale (a), and over East North America (b), West and Central Europe (c), the Mediterranean (d), East Asia (e), and South Asia (f). The boundaries of each region of interest are shown in the embedded map in subplot (a).

1415

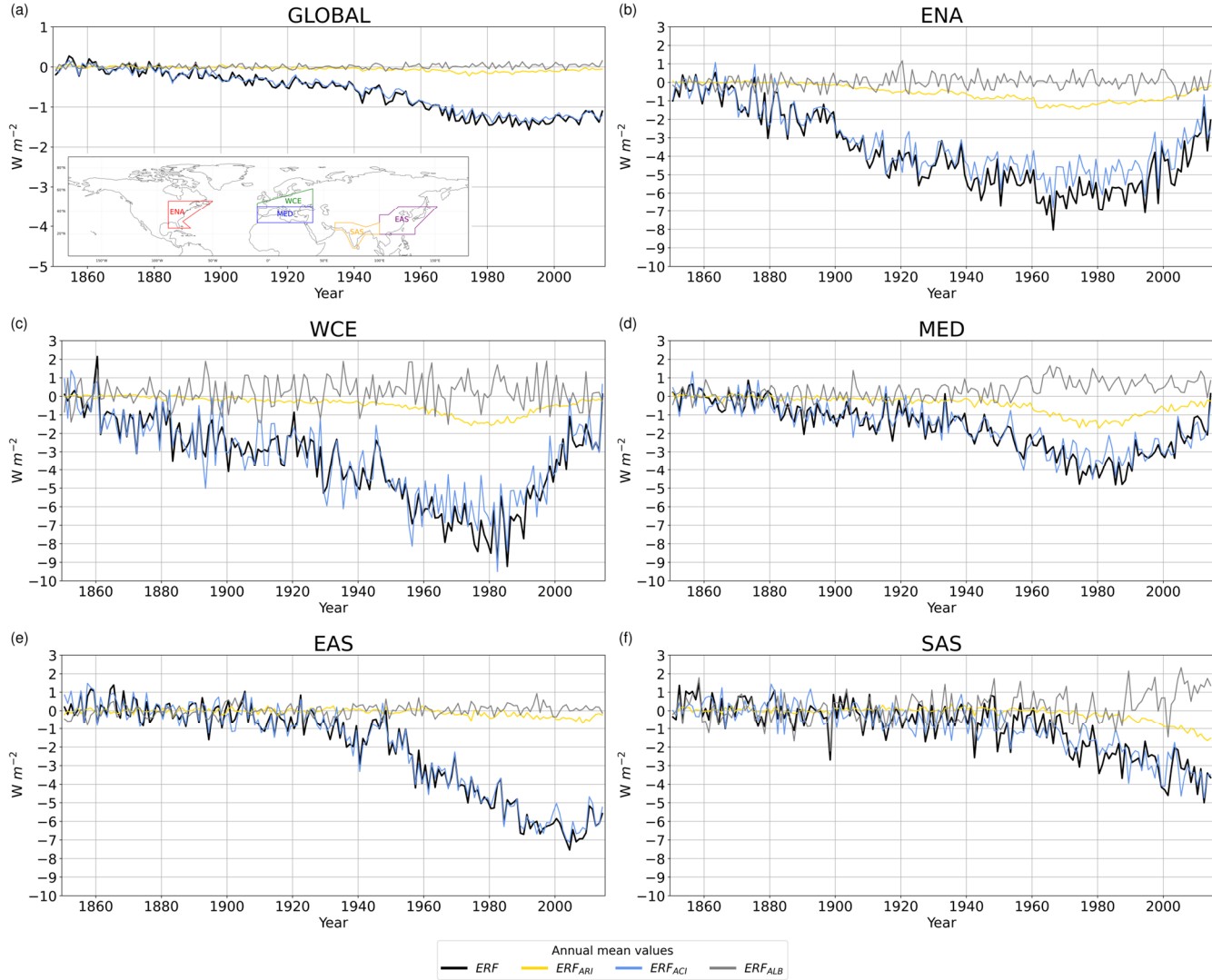

**Figure 9.** Time evolution of the total ERF, ERF$_{ARI}$, ERF$_{ACI}$, and ERF$_{ALB}$ due to anthropogenic aerosols over the historical period (1850-2014). The results are presented for the histSST experiment on global scale (a), and over East North America (b), West and Central Europe (c), the Mediterranean (d), East Asia (e), and South Asia (f). The boundaries of each region are shown in the embedded map in subplot (a).

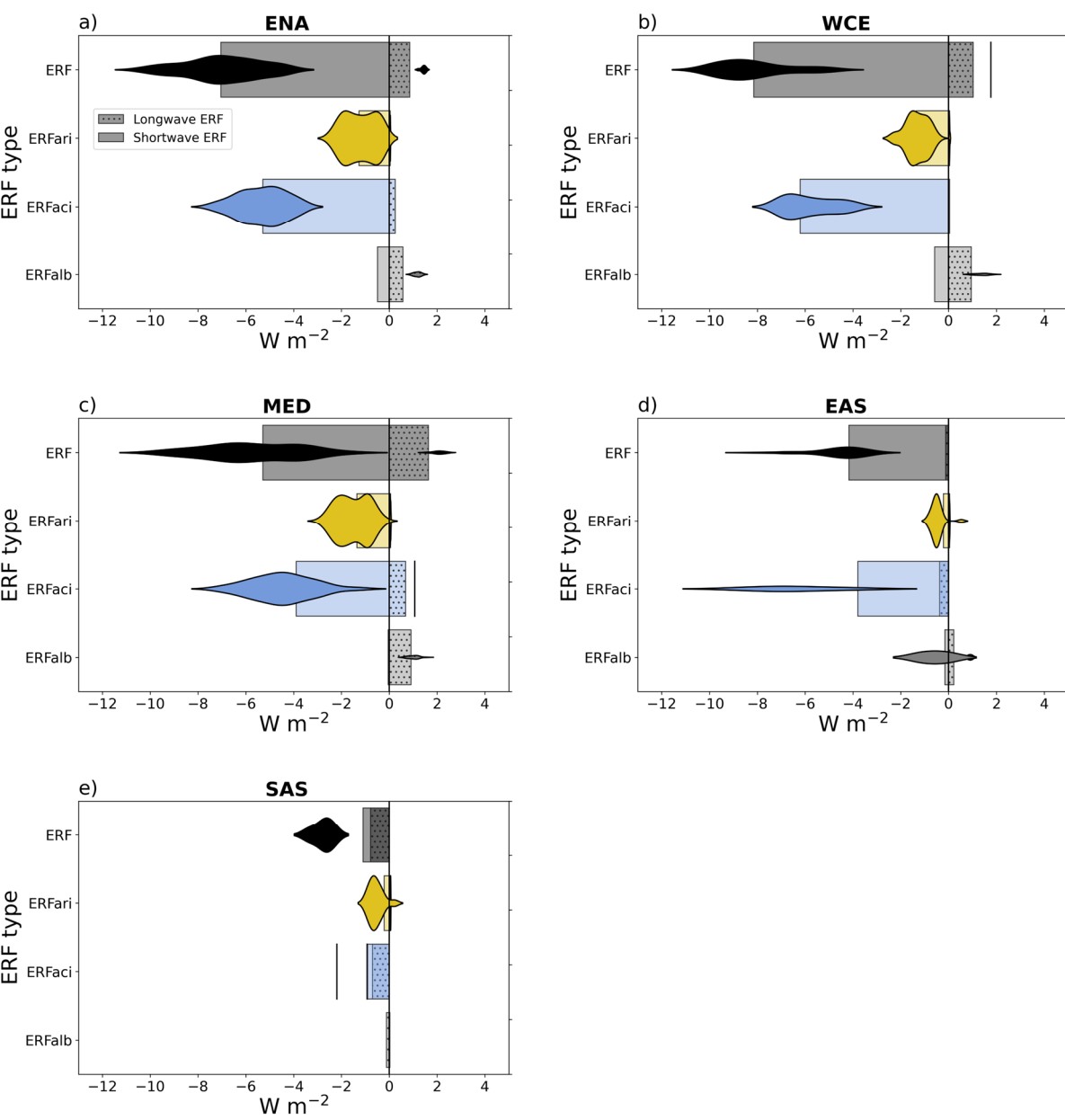

**Figure 10.** SW and LW decomposition of ERF over East North America (a), West and Central Europe (b), the Mediterranean (c), East Asia (d), and South Asia (e). The violins show the distribution of values over regions where ERFs are statistically significant.

1425

**SW and LW ERF by Region (histSST - histSST-piAer) 1995-2014**

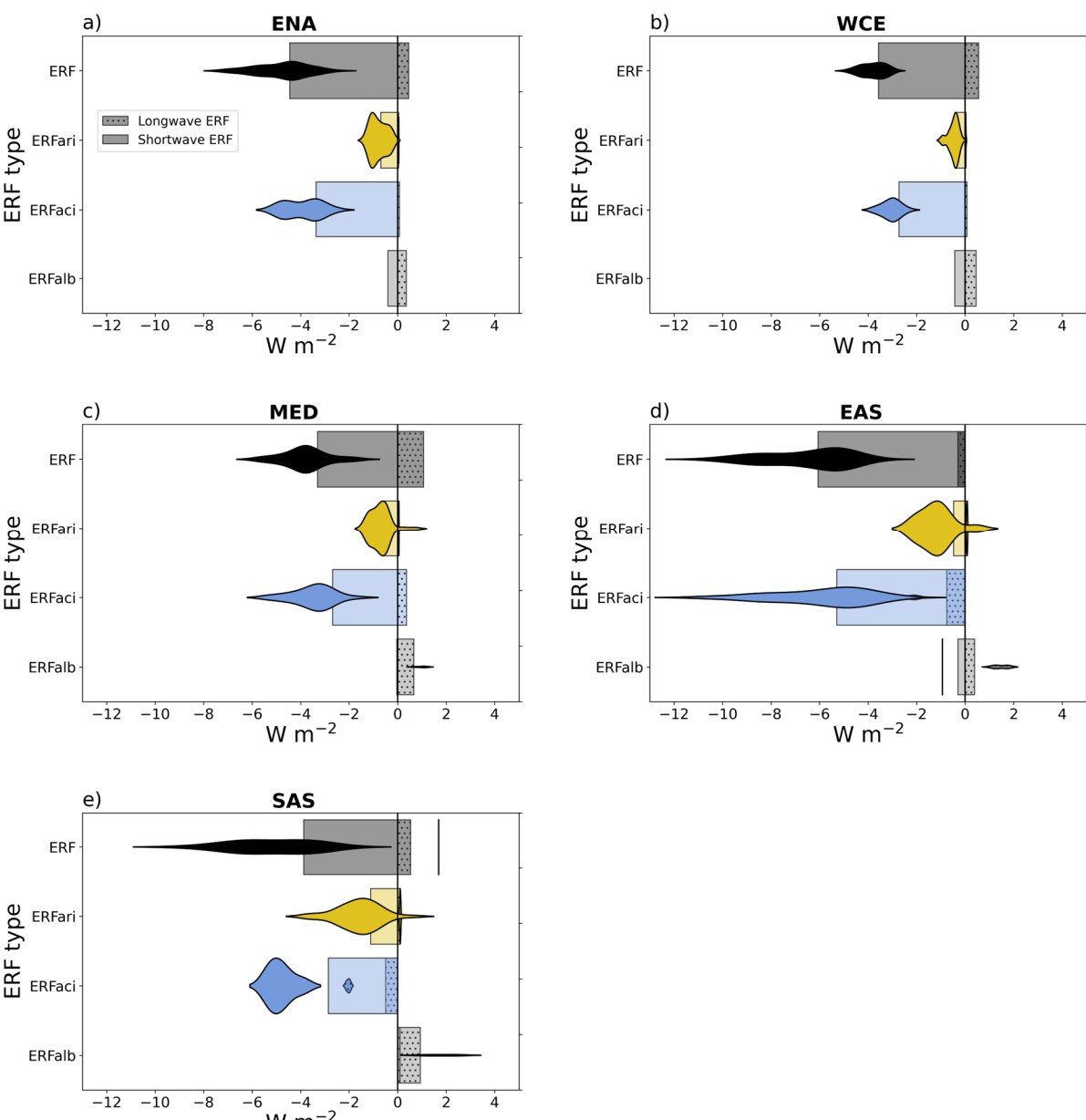

1430     **Figure 11.** As in Fig. 10, but for the 1995-2014 period.