# Peer review of "Decomposing the Effective Radiative Forcing of anthropogenic aerosols based on CMIP6 Earth System Models"

_EGUsphere, 2023_

## Author Response (AR1)

**ARISTOTLE UNIVERSITY OF THESSALONIKI**
**SCHOOL OF GEOLOGY**

**DEPARTMENT OF METEOROLOGY AND**
**CLIMATOLOGY**

*School of Geology*
*541 24 – Thessaloniki*
*Greece*

*Tel: +30 2310-998480*
*e-mail: kalisort@geo.auth.gr*

17 February 2024

Dear Editor

We have submitted our revised manuscript with title "*Decomposing the Effective Radiative Forcing of anthropogenic aerosols based on CMIP6 Earth System Models*" for potential publication in Atmospheric Chemistry and Physics. We considered all the comments of the reviewers and there is a detailed response on their comments point by point (see below). We believe that our study represents a significant contribution in our understanding of the effective radiative forcing (ERF) induced by anthropogenic aerosols on global and regional scale, and we hope that you will consider our manuscript for publication.

Yours sincerely,

Alkiviadis Kalisoras
Ph.D. Candidate

**Reply to Reviewer #1**

We would like to thank Reviewer #1 for the constructive and helpful comments. The reviewer's contribution is recognized in the acknowledgments of the revised manuscript. Below follows our response point by point. The reviewer's comments are given in *italic* and our response is given in **bold** font.

General comments:

*1)* The Reviewer notes: "*It would be good to briefly introduce which input datasets (e.g. emission data sets) and parameterizations are used in these models, which are highly relevant with ERF and could contribute to the inter-model uncertainties, such as cloud microphysical schemes (one moment/two moment, bulk/bin), activation, autoconversion schemes?*"

**Information regarding emission datasets and model parameterizations was added in Section 2 of the manuscript as a separate subsection (Subsection 2.1 Models Description).**

*2)* The Reviewer notes: "*Semi-direct effects have been mentioned in the introduction part, but not discussed enough in the main text. These effects are especially important in interpreting ERF from BC aerosols, and could largely contribute to inter-model uncertainties. Clarification on whether these effects are included in the ERF calculations and how the models' parameterizations impact these uncertainties would enhance the study's comprehensiveness.*"

**The method of Ghan (2013) cannot explicitly calculate the magnitude of semi-direct effects. The ERFaci term is an estimate of anthropogenic aerosol effects on cloud radiative forcing, which is the sum of aerosol indirect effects and semi-direct effects. Therefore, Ghan's decomposition cannot isolate the semi-direct effects as it would require additional diagnostics (Ghan et al., 2012; Ghan, 2013; Zelinka et al., 2023). This information has also been added to Subsection 2.3 Methodology.**

Minor comments:

*3)* The Reviewer notes: "*L31: the range showing one standard deviation?*"

**Yes, the ranges show one standard deviation, which was calculated using the area-weighted field mean values of the corresponding ESMs.**

*4)* The Reviewer notes: "*L39: ERF from anthropogenic aerosols?*"

**Yes, it refers to the ERF from anthropogenic aerosols. The word "anthropogenic" was added in front of the word "aerosols".**

*5)* The Reviewer notes: "*L45: spatially heterogeneously*"

**It was revised accordingly as suggested by the reviewer.**

*6)* The Reviewer notes: "*L53: Not all types of aerosols can 'efficiently' serve as CCN or IN.... It relies on sizes, types, supersaturation, mixing state, …*"

**The reviewer is correct and, therefore, "efficiently" was replaced by "can".**

*7)* The Reviewer notes: "*L67: I would prefer 'conditions' than 'parameters' here.*"

It was revised accordingly as suggested by the reviewer. This sentence was rephrased as follows: "Nevertheless, in a more general sense, the term semi-direct effect can be used to express the thermodynamic effect of absorbing aerosols on meteorological conditions (atmospheric pressure, temperature profile and cloudiness, etc.) (Tsikerdekis et al., 2019).".

*8)* The Reviewer notes: "*L74: suggest also cite Martin Wild's dimming effect paper here.*"

The references of Wild (2009, 2012) were added in this sentence.

*9)* The Reviewer notes: "*L105: Additionally, the magnitude of ERFaci might also depend on dynamic backgrounds (Zhang et al., 2016; 10.5194/acp-16-2765-2016) and large-scale circulation adjustments (Dagan et al., 2023: https://doi.org/10.1038/s41561-023-01319-8).*"

The reviewer is right. The following sentence was added: "Moreover, the magnitude of the radiative forcing due to ACI could also depend on dynamic backgrounds (Zhang et al., 2016) as well as large-scale circulation adjustments (Dagan et al., 2023)."

*10)* The Reviewer notes: "*L105: 'on aerosol radiative forcing calculations': a work by Ghan et al., (2016) (10.1073/pnas.1514036113) might also be relevant, which demonstrates the chain processes within ERFaer and discussed the uncertainties of each process in GCMs.*"

The reference of Ghan et al. (2016) was added to the sentence.

*11)* The Reviewer notes: "*L125: 'but this would be difficult to apply in some climate models (Ramaswamy et al., 2019).': some recent work has done this by fixing land surface temperature, see Andrews et al, 2021: https://doi.org/10.1029/2020JD033880)*"

The reviewer is right. However, Lines 110-130 were omitted to enhance the readability of the paper.

*12)* The Reviewer notes: "*L131: ''The total ERF due to aerosols' : anthropogenic aerosols?*"

Yes, the ERF caused by anthropogenic aerosols is discussed in this sentence. The word "anthropogenic" was added in front of the word "aerosols".

*13)* The Reviewer notes: "*L142: The current paragraph appears to be overly dense with information, much of which seems to be a repetition of what is already presented in Table 1. It would be beneficial for the readers, in terms of enhanced readability and comprehension, if the key points and implications of these data were more clearly and explicitly given.*"

The reviewer is right. As this discussion is also summarized in Table 1, this whole paragraph was omitted to enhance the paper's readability.

*14)* The Reviewer notes: "*L275: Here and other places, please add units*"

Changes in AOD (which are unitless) are investigated using four CMIP6 variables (i.e., od550aer, od550so4, od550oa and od550bc).

*15)* The Reviewer notes: "*L289: Are stratospheric (volcanic) aerosols included in od550so4?*"

The variable "od550so4" corresponds to atmosphere optical thickness due to sulphate ambient aerosol particles. It does not include AOD from stratospheric aerosols if these are prescribed but includes other possible background aerosol types. In most models the stratospheric aerosols are prescribed (by their SW and LW radiative properties), and "od550so4" only includes the contribution from tropospheric aerosols. The same applies to "od550aer", "od550bc", and "od550oa" that were also used in our study.

*16)* The Reviewer notes: "*L316-318: Positive ERFari over these regions are mostly due to absorbing aerosols?*"

Yes, the positive ERFari due to aerosols over Central Africa, the Arabian Desert and continental East Asia (Fig. 2c, d) can be attributed to absorbing aerosols and in particular anthropogenic black carbon (as opposed to dust). This is also supported by the fact that peak positive ERFari values due to black carbon are found over the same regions (Fig. 3f).

*17)* The Reviewer notes: "*L398: How does albedo change the LW ERF? Isn't it primarily influencing shortwave radiation by changing how solar energy is absorbed or reflected?*"

Ghan (2013) states that ERFalb term "includes effects of both changes in snow albedo due to deposition of absorbing aerosol, and changes in snow cover induced by deposition and by the other aerosol forcing mechanisms". However, this refers only to SW ERFalb, which also includes "the aerosol-free clear-sky radiative contributions from changes in humidity plus a masking term that quantifies how much the radiative impact of changes in surface albedo is attenuated by the presence of both clouds and aerosols" (Zelinka et al., 2023). The LW component of this forcing term, which we name "ERFalb" in our manuscript only for compatibility purposes with the respective term used in the paper of Ghan (2013), includes the aerosol-free clear-sky radiative contributions from changes in temperature and humidity (Zelinka et al., 2023). This information has also been added to Subsection 2.3 Methodology.

*18)* The Reviewer notes: "*L399: 'borne to mind' change to 'borne in mind'*"

It was revised accordingly as suggested by the reviewer.

*19)* The Reviewer notes: "*L422: From Fig. 5, for BC, ERF LW is still positive, ERFari LW around zero and ERFaci LW positive - I didn't see 'a negative but weaker LW ERF' from BC...*"

This sentence refers to the SW and LW ERF values from individual models presented in Tables S2 and S3, and not to the ensemble mean, which is presented in Fig. 5. We realize that this sentence might be confusing, so we revised it as follows: "Nearly all individual models produce a positive total BC ERF (Table 4) arising from the positive SW ERF due to absorption of solar incoming radiation (Table S2), which is offset by a negative, but weaker, LW ERF (Table S3)."

*20)* The Reviewer notes: "*L438: For this paragraph which focused on BC, it would benefit from some discussions on semi-direct effect.*"

The reviewer is right. However, as the semi-direct effect cannot be explicitly quantified using the method of Ghan (2013), it is not further discussed in this paragraph. This paragraph was dedicated to the discussion of SW and LW ERF due to black carbon, which was calculated using the Ghan (2013) decomposition.

*21)* The Reviewer notes: "*Table 5: Briefly introduce what the abbreviations stand for - captions should be self-explained.*"

It was revised accordingly as suggested by the reviewer.

*22)* The Reviewer notes: "*Fig 1,2,3: It is really hard to tell the difference between '///' and 'xx' symbols in the figures…*"

**While we bolded the 'xx' symbols in order to make them more distinct, we realize that both symbols are quite small. Our main goal was to qualitatively distinguish the robust from the non-robust results in Figs. 1-3 and not to focus on the quantitative differences of the non-robust results (method used for the robustness of the results is summarized in Table A2). The statistical significance of the results for each model can also be derived from Figs. S3, S4, S7, S8 and S9, where the '+' symbol denotes the statistical significance. We will collaborate closely with the production team of the journal to make sure that all figures appear large enough in the final paper so that anyone can distinguish the differences between symbols.**

*23)* The Reviewer notes: "*Fig 7: I like the idea of showing the relative importance of ACI, ARI, and ALB geographically. Could you explain why there are some regions dominated by ALB over ocean in the BC case?*"

**As explained in Comment #17 ERFalb is influenced by more than aerosol-induced changes in surface albedo (Zelinka et al., 2023). Based on equations (16), (B18) and (B20) in the paper of Zelinka et al. (2023), the total (i.e., SW+LW) ERFalb, which was used in our method, equals the change in net radiation due to surface albedo changes plus the aerosol-free clear-sky radiative contributions from changes in both temperature and humidity plus a masking term that quantifies how much the radiative impact of changes in surface albedo is attenuated by the presence of both clouds and aerosols. The method we used to determine the dominant ERF component is too simplistic to provide accurate insights to the underlying physical processes that lead to the forcing. All we can safely comment is that ERFalb dominates the ARI and ACI terms over these regions.**

**Reply to Reviewer #2**

We would like to thank Reviewer #2 for the constructive and helpful comments. The reviewer's contribution is recognized in the acknowledgments of the revised manuscript. Below follows our response point by point. The reviewer's comments are given in *italic* and our response is given in **bold** font.

Major comments:

*1)* The Reviewer notes: "*1. The authors should more strongly emphasize novel aspects in their introduction, discussion, and conclusions and shorten the discussion of more general aspects. Some of the general discussions are rather lengthy while still missing essential points. These general discussions should be shortened.*"

**The purpose and novelties of this paper are highlighted throughout the manuscript (e.g., Lines 175-184, Lines 218-220, Lines 265-267 and Lines 440-446). The novelty of this paper lies in the investigation of more technical aspects of ERF estimation, such as the robustness of the results, the relative contribution of ERFari, ERFaci and ERFalb to the total ERF, and the temporal evolution of ERFari, ERFaci and ERFalb in both global and regional scale (also see reply to Comment #5). This study reviews and complements the findings of previous studies by providing figures with ΔAOD and ERF spatial patterns and tables with weighted mean values and standard deviations on global and regional scale. Yet, indeed some parts of the introduction are lengthy, and are shortened in the revised manuscript as suggested.**

*2)* The Reviewer notes: "*2. Thornhill et al. (2021) present a number of very similar results. I think that in order to justify another publication on this topic, the authors should try to come up with additional findings. Specifically, I suggest to analyze which regions contribute most strongly to the spread of model results for global mean ERFaci. I suggest to use a pragmatic separation into ΔAOD regimes based on multi-model average AOD change (ΔAOD) in Figures 1 a and b. For example, the authors could distinguish between high to medium ΔAOD source regions over land, medium to low ΔAOD regions over land, high ΔAOD over ocean in the outflow and low ΔAOD over ocean (remote ocean). I think it would be interesting not only to specify ERFaci standard deviations in W m⁻² but also as contributions to uncertainty in global mean ERFaci, which involves area weighting.*"

**Analysis of the ERF inter-model variability (one standard deviation) indicates that ERFaci is the main source of uncertainty in total ERF (Table I, Fig. I). East Asia contributes the most to the inter-model spread of both ΔAOD (Table II) and ERFaci results with the standard deviation of the latter exceeding 5.5 W m⁻². Tables I and II were incorporated inTables 5 and S5, respectively. Figure I was also added to the Supplement.**

**Table I. Inter-model variability (one standard deviation) of ERF (in W m⁻²) during the negative ERF peak period (1965-1984) and the recent past (1995-2014) from the histSST experiment. Global and regional ERF standard deviations are presented for the five regions investigated in the paper: East North America (ENA), West and Central Europe (WCE), the Mediterranean (MED), East Asia (EAS) and South Asia (SAS).**

| Region | 1965-1984 | | | | 1995-2014 | | | |
|--------|-----|-----|-----|-----|-----|-----|-----|-----|
|        | ERF | ARI | ACI | ALB | ERF | ARI | ACI | ALB |
| ENA    | 1.61 | 0.59 | 1.84 | 0.15 | 1.00 | 0.39 | 1.23 | 0.15 |
| WCE    | 3.00 | 0.59 | 3.65 | 0.51 | 1.13 | 0.24 | 1.35 | 0.49 |
| MED    | 0.96 | 0.45 | 1.40 | 0.30 | 0.61 | 0.31 | 1.18 | 0.52 |
| EAS    | 2.45 | 0.46 | 2.90 | 0.25 | 3.06 | 1.03 | 3.71 | 0.18 |
| SAS    | 0.89 | 0.31 | 0.97 | 0.46 | 0.91 | 0.86 | 1.76 | 0.92 |
| GLOBAL | 0.43 | 0.11 | 0.44 | 0.05 | 0.37 | 0.14 | 0.44 | 0.08 |

**Intermodel Variability of ERF**

[Figure]

**Figure I: Inter-model variability (one standard deviation) of total (SW+LW) ERF (a, b), ERFari (c, d), ERFaci (e, f) and ERFalb (g, h) due to all anthropogenic aerosols relative to the pre-industrial era. The spatial distribution is presented for the multi-model ensembles of piClim-aer (left column) and histSST (averaged over 1995-2014; right column) experiments, respectively.**

**Table II.** Inter-model variability (one standard deviation) of ΔAOD for histSST experiment averaged over 1965-1984 and 1995-2014. Variables od550aer, od550so4, od550oa, and od550bc denote the differences in all-aerosol, sulphate, organic aerosol, and black carbon AOD, respectively. Global and regional ΔAOD standard deviations are presented for the five regions investigated in the paper: East North America (ENA), West and Central Europe (WCE), the Mediterranean (MED), East Asia (EAS) and South Asia (SAS).

| Region | 1965-1984 | | | | 1995-2014 | | | |
|---|---|---|---|---|---|---|---|---|
| | od550aer | od550so4 | od550oa | od550bc | od550aer | od550so4 | od550oa | od550bc |
| ENA | 0.0418 | 0.0224 | 0.0095 | 0.0006 | 0.0204 | 0.0168 | 0.0080 | 0.0006 |
| WCE | 0.1182 | 0.0878 | 0.0076 | 0.0015 | 0.0310 | 0.0276 | 0.0046 | 0.0008 |
| MED | 0.0258 | 0.0191 | 0.0035 | 0.0005 | 0.0162 | 0.0129 | 0.0033 | 0.0006 |
| EAS | 0.0340 | 0.0262 | 0.0086 | 0.0017 | 0.0839 | 0.0649 | 0.0216 | 0.0039 |
| SAS | 0.0127 | 0.0085 | 0.0069 | 0.0008 | 0.0495 | 0.0309 | 0.0187 | 0.0015 |
| GLOBAL | 0.0065 | 0.0068 | 0.0023 | 0.0003 | 0.0088 | 0.0077 | 0.0039 | 0.0005 |

**Based on Fig. 1a and 1b of the manuscript, four ΔAOD regimes can be distinguished:**

a) **High to medium ΔAOD over land: East and South Asia, Eastern Europe, Middle East, Eastern North America**

b) **Medium to low ΔAOD over land: North and South America, Western Europe, Greenland, Oceania, Antarctica, Arctic**

c) **High ΔAOD over ocean: Northwestern Pacific, Northernmost Indian**

d) **Low ΔAOD over ocean: Atlantic, South Pacific, South Indian**

**The above ΔAOD regime discussion was added to the "Results" section.**

*3)* The Reviewer notes: "*3. It is widely understood that emissions took different trajectories in Europe and Asia and also that the trajectories for India and China have diverged. I think the authors should either shorten or omit the analysis of selected regions or else explain better what motivated this part of the study and what is new or unexpected about their results.*"

**The time evolution of ΔAOD and all ERF components (ARI, ACI and ALB) is shown on global scale and over ERF hotspots during 1850-2014 using histSST experiments. The selected regions are highly industrialized regions that exert the most negative ERF values during the historical period. Weighted field means of ΔAOD and ERF over the selected regions are also presented for every ESM and their ensemble for two time periods of interest: i) during the negative ERF peak (1965-1984; see also Szopa et al., 2021) and ii) the end of historical simulations (1995-2014). Moreover, SW and LW ERF values are presented over the selected regions for both time periods of interest, which not only addresses the errata of IPCC AR6 Chapter 6 (Szopa et al., 2021; https://www.ipcc.ch/report/ar6/wg1/downloads/report/IPCC_AR6_WGI_Errata.pdf), but also attempts to explain the underlying physical processes by studying changes in liquid and ice water paths (Fig. S10 in Supplement).**

*4)* The Reviewer notes: "*4. Lines 537-556: I think that in the conclusion section the authors should summarize results and draw conclusions instead of spending an entire long paragraph simply repeating what they did.*"

**The reviewer is right. As a result, the "Conclusions" section was modified (see reply to Comment #5).**

*5)* The Reviewer notes: "*5. Lines 536-611: Novel findings should be highlighted. If the only main points are to demonstrate that the authors used similar methods to arrive at similar results compared to previous studies, then I do not understand why this study presents an advance over previous studies and should be published.*"

**The authors of this paper strongly believe that this study presents a number of novel findings as it is a comprehensive spatiotemporal ERF analysis, which complements and advances the findings of other papers:**

a) A concise ensemble of seven CMIP6 Earth System models was used to calculate the ERF of anthropogenic aerosols (as a whole and for different sub-species, such as black and organic carbon, and sulphates) from two different sets of experiments (piClim series and histSST) based on the method of Ghan (2013), which is considered a very accurate method (Zelinka et al., 2014; Michou et al., 2020).

b) Spatial patterns at top-of-atmosphere and global weighted field means for all SW, LW and total (i.e., SW+LW) ERF components (ARI, ACI and ALB) are presented for every ESM and their ensemble and for every experiment (piClim-aer, piClim-BC, piClim-OC, piClim-SO$_2$ and histSST). Inter-model variability (one standard deviation) is also shown in the case of the ensemble. To our knowledge, the information obtained from a) and b) has not been presented all together in one paper.

c) The inter-model agreement as well as the robustness of ERF and ΔAOD ensemble results were calculated using a method similar to the one used in the 6[th] IPCC Assessment Report (described within the manuscript) and are shown in Fig. 1-3. This method of calculating the robustness of the results has not been used in other papers as far as we know.

d) The novel concept of determining the driving factor of ERF (ARI, ACI or ALB) on global scale is presented in Fig. 6-7 using a method described in detail in the text. This has not been done in other studies.

Although the novelty of this study is highlighted in the last paragraph of the introduction, the reviewer has a point. Considering Comments #4 and #5, the first paragraph in the "Conclusions" section was modified as follows: "In this work, the effective radiative forcing (ERF) of anthropogenic aerosols was investigated using fixed-SST simulations from seven different ESMs participating in the CMIP6 exercise. Shortwave (SW), longwave (LW) and total (i.e., SW+LW) ERF and changes in aerosol optical depth (AOD) were quantified for all anthropogenic aerosols, combined and individually, using both piClim and histSST experiments for comparison purposes. Additionally, the robustness of the multi-model ensemble results was calculated by investigating both the statistical significance of each model's results and the agreement between individual models on the sign of change. Spatial patterns and temporal evolution of ERF and ΔAOD were presented on global and regional scale, along with tables that show the area-weighted mean values and standard deviation of ERF and ΔAOD for the multi-model ensemble as well as every individual model."

Specific comments:

*6) The Reviewer notes: "Line 23: I think that "which is the recommended metric for perturbations affecting the Earth's top-of-atmosphere energy budget since it is a better way to link this perturbation to subsequent global mean surface temperature change" and also the corresponding lengthy discussion in lines 106 to 130 of the introduction should be omitted. The question whether ERF is a good metric is not addressed by the results of this study."*

The reviewer is correct. Lines 23-24 and Lines 110-130 were omitted.

*7) The Reviewer notes: "Line 62: "can be observed" -> please elaborate"*

This sentence merely addresses the fact that a semi-direct effect of aerosols exists and has been investigated in a number of papers mentioned in the manuscript. Although it is not investigated individually, it is part of the ERFaci term of Ghan's (2013) decomposition and for reasons of completeness, it is discussed in the introduction.

*8) The Reviewer notes: "Line 214: I think that for an ERF it is sufficient that identical SST and SIC are prescribed in the base and the perturbed run. The (first order) question is whether SST and SIC are allowed to respond to the forcing."*

The histSST and histSST-piAer simulations use atmosphere-only configurations with prescribed sea-surface temperatures and sea ice (Collins et al., 2017). Therefore, SSTs and SIC are not allowed to respond to the aerosol forcing.

*9) The Reviewer notes: "Lines 218-220: Perhaps explain and motivate this in the introduction?"*

Explaining this in the introduction would be confusing for the readers, as the simulations used in this paper are described in Chapter 2. However, the following sentences were added to the last paragraph of the introduction: "The present-day anthropogenic aerosol ERF is examined at the top-of-the-atmosphere using two different sets of experiments with fixed sea surface temperatures (SSTs) and sea ice cover (SIC) for comparison purposes. Moreover, the evolution of transient ERF during the historical period (1850-2014) is investigated globally and over certain emission regions of the Northern Hemisphere (NH), focusing on the last 20 years of the historical period (1995-2014) in order to mitigate the effects of the negative ERF peak around in late 1970s (Szopa et al., 2021)."

10) The Reviewer notes: "*Lines 378-381: What do we learn from this? If I interpret it correctly, Figure 4 suggests that the values are consistent.*"

Lines 378-381 show the similarities between piClim-aer and histSST (averaged over 1995-2014), but also highlight the differences between ESMs in ERFari and ERFaci. While all models agree on the negative sign of ERFaci for both experiments, there are discrepancies in the sign of ERFari. These statements were also added in the revised manuscript.

11) The Reviewer notes: "*Lines 454-455: Why?*"

Due to differences in experimental set-up between piClim-aer and piClim-SO₂, as well as differences in the number of models used for the ERF decomposition (EC-Earth3-AerChem did not participate in the calculation of ERF for piClim-SO₂, piClim-OC and piClim-BC due lack of diagnostics as stated in Lines 262-263). One possible reason is that some of the cooling by sulphate aerosols is compensated by warming by BC in the piClim-aer experiment compared to piClim-SO₂. However, it should also be noted that, while the method used to determine the dominant ERF component can provide some insights, it is quite simplistic and, therefore, cannot fully explain the underlying physical processes that lead to the forcing.

12) The Reviewer notes: "*Table 4: For MPI-ESM-1-2-HAM piClim, the sum of ERFs for individual species seems to differ more from the total ERF than for other models. Do you have idea why this could be?*"

MPI-ESM-1-2-HAM produces a highly negative ERFaci in both piClim-OC and piClim-BC, thus leading to more negative total ERF for both experiments. As a result, the sum of ERFs for individual aerosol species differs more from the ERF due to all anthropogenic aerosols. The reason behind these results is that coating by anthropogenic sulphate removes BC and OC in piClim-aer, whereas OC and BC have longer lifetimes in piClim-OC and piClim-BC respectively (not shown). Therefore, OC and BC are transported further and contribute more to ERF in piClim-OC and piClim-BC respectively than in piClim-aer.

Technical comments:

13) The Reviewer notes: "*Line 399: borne to mind -> borne in mind?*"

It was revised accordingly as suggested by the reviewer.

14) The Reviewer notes: "*Lines 399-400: "affecting the realizations and parameterization schemes ESMs use to quantify the magnitude of different processes" sound confusing. Please omit or rephrase. Avoid unnecessary repetitions.*"

Lines 398-401 were rephrased as follows: "It should be borne in mind that not all ESMs agree on the magnitude or even the sign of the individual SW and LW ERF main components (Tables S2-S4) due to uncertainties in the parameterization schemes used in ESMs to describe the way aerosols interact with radiation and clouds."

*15)* The Reviewer notes: "*Line 405: It is interesting to note that -> The*"

**It was revised accordingly as suggested by the reviewer.**

*16)* The Reviewer notes: "*Line 454: larger -> a larger*"

**It was revised accordingly as suggested by the reviewer.**

*17)* The Reviewer notes: "*Line 498: gets more positive values -> becomes less negative*"

**It was revised accordingly as suggested by the reviewer.**

**References**

Collins, W. J., Lamarque, J.-F., Schulz, M., Boucher, O., Eyring, V., Hegglin, M. I., Maycock, A., Myhre, G., Prather, M., Shindell, D., and Smith, S. J.: AerChemMIP: quantifying the effects of chemistry and aerosols in CMIP6, Geosci. Model Dev., 10, 585–607, https://doi.org/10.5194/gmd-10-585-2017, 2017.

Dagan, G., Yeheskel, N., and Williams, A. I. L.: Radiative forcing from aerosol–cloud interactions enhanced by large-scale circulation adjustments, Nat. Geosci., 16, 1092–1098, https://doi.org/10.1038/s41561-023-01319-8, 2023.

Ghan, S., Wang, M., Zhang, S., Ferrachat, S., Gettelman, A., Griesfeller, J., Kipling, Z., Lohmann, U., Morrison, H., Neubauer, D., Partridge, D. G., Stier, P., Takemura, T., Wang, H., and Zhang, K.: Challenges in constraining anthropogenic aerosol effects on cloud radiative forcing using present-day spatiotemporal variability, Proceedings of the National Academy of Sciences, 113, 5804–5811, https://doi.org/10.1073/pnas.1514036113, 2016.

Ghan, S. J.: Technical Note: Estimating aerosol effects on cloud radiative forcing, Atmos. Chem. Phys., 13, 9971–9974, https://doi.org/10.5194/acp-13-9971-2013, 2013.

Ghan, S. J., Liu, X., Easter, R. C., Zaveri, R., Rasch, P. J., Yoon, J.-H., and Eaton, B.: Toward a Minimal Representation of Aerosols in Climate Models: Comparative Decomposition of Aerosol Direct, Semidirect, and Indirect Radiative Forcing, Journal of Climate, 25, 6461–6476, https://doi.org/10.1175/JCLI-D-11-00650.1, 2012.

Michou, M., Nabat, P., Saint-Martin, D., Bock, J., Decharme, B., Mallet, M., Roehrig, R., Séférian, R., Sénési, S., and Voldoire, A.: Present-Day and Historical Aerosol and Ozone Characteristics in CNRM CMIP6 Simulations, J. Adv. Model. Earth Syst., 12, https://doi.org/10.1029/2019MS001816, 2020.

Szopa, S., Naik, V., Adhikary, B., Artaxo Netto, P. E., Berntsen, T., Collins, W. D., Fuzzi, S., Gallardo, L., Kiendler-Scharr, A., Klimont, Z., Liao, H., Unger, N., and Zanis, P.: Short-lived climate forcers, in: Climate Change 2021: The Physical Science Basis. Contribution of Working Group I to the Sixth Assessment Report of the Intergovernmental Panel on Climate Change, edited by: Masson-Delmotte, V., Zhai, P., Pirani, A., Connors, S. L., Péan, C., Berger, S., Caud, N., Chen, Y., Goldfarb, L., Gomis, M. I., Huang, M., Leitzell, K., Lonnoy, E., Matthews, J. B. R., Maycock, T. K., Waterfield, T., Yelekçi, Ö., Yu, R., and Zhou, B., Cambridge University Press, Cambridge, United Kingdom and New York, NY, USA, 817–922, https://doi.org/10.1017/9781009157896.001, 2021.

Tsikerdekis, A., Zanis, P., Georgoulias, A. K., Alexandri, G., Katragkou, E., Karacostas, T., and Solmon, F.: Direct and semi-direct radiative effect of North African dust in present and future regional climate simulations, Clim Dyn, 53, 4311–4336, https://doi.org/10.1007/s00382-019-04788-z, 2019.

Wild, M.: Global dimming and brightening: A review, Journal of Geophysical Research: Atmospheres, 114, https://doi.org/10.1029/2008JD011470, 2009.

Wild, M.: Enlightening Global Dimming and Brightening, Bulletin of the American Meteorological Society, 93, 27–37, https://doi.org/10.1175/BAMS-D-11-00074.1, 2012.

Zelinka, M. D., Andrews, T., Forster, P. M., and Taylor, K. E.: Quantifying components of aerosol-cloud-radiation interactions in climate models, J. Geophys. Res. Atmos., 119, 7599–7615, https://doi.org/10.1002/2014JD021710, 2014.

Zelinka, M. D., Smith, C. J., Qin, Y., and Taylor, K. E.: Comparison of methods to estimate aerosol effective radiative forcings in climate models, Atmospheric Chemistry and Physics, 23, 8879–8898, https://doi.org/10.5194/acp-23-8879-2023, 2023.

Zhang, S., Wang, M., Ghan, S. J., Ding, A., Wang, H., Zhang, K., Neubauer, D., Lohmann, U., Ferrachat, S., Takeamura, T., Gettelman, A., Morrison, H., Lee, Y., Shindell, D. T., Partridge, D. G., Stier, P., Kipling, Z., and Fu, C.: On the characteristics of aerosol indirect effect based on dynamic regimes in global climate models, Atmospheric Chemistry and Physics, 16, 2765–2783, https://doi.org/10.5194/acp-16-2765-2016,  2016.

---

## Author Response (AR2)

**Reply to Reviewer #2**

We would like to thank Reviewer #2 for the additional useful comments. Below follows our response point by point. The reviewer's comments are given in *italic* and our response is given in **bold** font.

*1)* The Reviewer notes: *"The authors did not address my main criticism. The abstract still fails to mention novel aspects. In the abstract, the authors should already make clear which results essentially stem from repeating other studies and which results are new."*

**The abstract has been restructured in order to address the reviewer's comment to highlight the key findings and novelty of this work.**

*2)* The Reviewer notes: *"Looking at contributions of uncertainties from different regions, as far as I can see, the authors focused on 'hotspots'. I wonder how much different regions - including remote oceanic regions - contribute to model spread in ERFaci. Is the spread caused only by the hotspots? How much do remote regions contribute to the spread between models?"*

**Following reviewer's comment, we added the following paragraph in Section 3.6 concluding that regions with high to medium ΔAOD over land tend to show larger inter-model variability in total ERF and total $ERF_{ACI}$ than land regions with medium to low ΔAOD, with the lowest inter-model spread appearing over remote oceanic regions with medium to low ΔAOD:**

[revised manuscript text omitted]

---

## Author Response (AR3)

**Reply to Reviewer #2**

We would like to thank Reviewer #2 for the additional useful comments. Below follows our response point by point. The reviewer's comments are given in *italic* and our response is given in **bold** font.

*1)* The Reviewer notes: *"I appreciate that the authors have re-written their abstract to better reflect novel aspects of their study. However, the newly added paragraph in Section 3.6 compares standard deviations between different regions. But this does not help to understand which region contributes most to uncertainty in total ERFaci. Quantifying which region contributes most to uncertainty in the global mean requires area weighting."*

**The standard deviations of different regions presented in Section 3.6 are area-weighted since they are calculated from the area-weighted mean regional ERFs of the individual models. In order to investigate which regions contribute most to the uncertainty of ERF (and its components), we first calculated the area-weighted standard deviation for a number of regions defined in Gutiérrez et al. (2021) from Fig. S17 and then estimated the percent contribution to uncertainty using the following formula:**

$$\frac{\sigma_R}{\sigma_G} \times \frac{A_R}{A_G} \times 100 \quad (1),$$

**where $\sigma_R$ and $\sigma_G$ are the area-weighted standard deviations of the region and the globe, respectively, and $A_R$ and $A_G$ are the surface areas of the region and the globe (after regridding – see Section 2.3), respectively. In Table I below, we show the percent contribution of all ERF components for several ATLAS regions of interest, along with their surface areas in $Km^2$. East Asia has the land region with the largest contribution to $ERF_{ARI}$ and $ERF_{ACI}$ uncertainty (9.15% and 6.53%, respectively) by far. There are also oceanic regions that largely contribute to $ERF_{ACI}$ uncertainty (such as the Central and East North Pacific Ocean), but this is due to their large surface areas affecting the result (Eq. 1). Areas not included in Table I have small contributions (< 4%) to the uncertainty of $ERF_{ARI}$, $ERF_{ACI}$, and $ERF_{ALB}$. This information was added to Section 3.6.**

**Table I. Contribution (in %) of the regional total ERFs to the global total ERF uncertainty for a number of ATLAS regions. The surface areas of each region are presented in Km².**

| Region | Surface Area | ERF | ARI | ACI | ALB |
|---|---|---|---|---|---|
| East North America | 5,761,900.38 | 1.64 | 2.02 | 1.77 | 1.05 |
| West and Central Europe | 3,717,508.80 | 0.89 | 0.82 | 1.02 | 1.18 |
| Mediterranean | 6,921,701.15 | 1.37 | 2.27 | 1.73 | 2.05 |
| Sahara | 11,005,305.48 | 2.21 | 4.47 | 1.21 | 3.54 |
| East Asia | 9,383,381.53 | 5.79 | 9.15 | 6.53 | 2.39 |
| Arabian Peninsula | 3,513,646.80 | 0.96 | 2.66 | 0.45 | 1.49 |
| South Asia | 5,677,956.74 | 1.85 | 4.89 | 1.98 | 3.18 |
| Southeast Asia | 17,171,955.47 | 4.31 | 5.13 | 4.06 | 3.50 |
| East Antarctica | 14,352,549.81 | 1.26 | 0.42 | 0.84 | 2.99 |
| West Antarctica | 5,395,830.36 | 0.38 | 0.15 | 0.43 | 1.09 |
| Arctic Ocean | 6,231,731.98 | 1.19 | 0.89 | 0.84 | 4.29 |
| West North Pacific Ocean | 18,639,975.87 | 5.73 | 3.61 | 5.88 | 2.07 |
| Central and East North Pacific Ocean | 31,138,651.29 | 8.55 | 5.03 | 8.76 | 3.75 |
| West South Pacific Ocean | 5,721,202.30 | 0.54 | 0.40 | 0.66 | 1.14 |
| Central and East South Pacific Ocean | 46,160,223.14 | 5.97 | 2.77 | 5.99 | 4.37 |
| North Atlantic Ocean | 23,927,204.38 | 5.87 | 5.77 | 6.43 | 2.95 |
| Southern Ocean | 51,367,264.97 | 6.83 | 2.54 | 6.61 | 3.16 |
| Global | 510,064,471.91 | 100.00 | 100.00 | 100.00 | 100.00 |

**References**

1) Gutiérrez, J. M., Jones, R. G., Narisma, G. T., Muniz Alves, L., Amjad, M., Gorodetskaya, I. V., Grose, M., Klutse, N. A. B., Krakovska, S., Li, J., Martínez-Castro, D., Mearns, L. O., Mernild, S. H., Ngo-Duc, T., van den Hurk, B., and Yoon, J.-H.: Atlas, in: Climate Change 2021: The Physical Science Basis. Contribution of Working Group I to the Sixth Assessment Report of the Intergovernmental Panel on Climate Change, edited by: Masson-Delmotte, V., Zhai, P., Pirani, A., Connors, S. L., Péan, C., Berger, S., Caud, N., Chen, Y., Goldfarb, L., Gomis, M. I., Huang, M., Leitzell, K., Lonnoy, E., Matthews, J. B. R., Maycock, T. K., Waterfield, T., Yelekçi, Ö., Yu, R., and Zhou, B., Cambridge University Press, Cambridge, United Kingdom and New York, NY, USA, 1927–2058, https://doi.org/10.1017/9781009157896.001, 2021.